# Fair and Useful Cohort Selection

**Konstantina Bairaktari**                                   *bairaktari.k@northeastern.edu*
*Khoury College of Computer Sciences*
*Northeastern University*

**Paul Langton**                                             *langton.p@northeastern.edu*
*Khoury College of Computer Sciences*
*Northeastern University*

**Huy Le Nguyen**                                            *hu.nguyen@northeastern.edu*
*Khoury College of Computer Sciences*
*Northeastern University*

**Niklas Smedemark-Margulies**                   *smedemark-margulie.n@northeastern.edu*
*Khoury College of Computer Sciences*
*Northeastern University*

**Jonathan Ullman**                                          *jullman@ccs.neu.edu*
*Khoury College of Computer Sciences*
*Northeastern University*

*Reviewed on OpenReview:* `https://openreview.net/forum?id=wRepWp1KC7`

## Abstract

A challenge in fair algorithm design is that, while there are compelling notions of individual fairness, these notions typically do not satisfy desirable composition properties, and downstream applications based on fair classifiers might not preserve fairness. To study fairness under composition, Dwork & Ilvento (2019) introduced an archetypal problem called *fair-cohort-selection problem*, where a single fair classifier is composed with itself to select a group of candidates of a given size, and proposed a solution to this problem. In this work we design algorithms for selecting cohorts that not only preserve fairness, but also maximize the utility of the selected cohort under two notions of utility that we introduce and motivate. We give optimal (or approximately optimal) polynomial-time algorithms for this problem in both an offline setting, and an online setting where candidates arrive one at a time and are classified as they arrive.

## 1 Introduction

The rise of algorithmic decision making has created a need for research on the design of fair algorithms, especially for machine-learning tasks like classification and prediction. Beginning with the seminal work of Dwork et al. (2012), there is now a large body of algorithms that preserve various notions of fairness for individuals. These notions of individual fairness capture the principle that similar people should be treated similarly, according to some task-specific measure of similarity.

While these algorithms satisfy compelling notions of individual fairness in isolation, they will often be used as parts of larger systems. To address this broader context, Dwork & Ilvento (2019) initiated the study of fairness under *composition*, where one or more fair classifiers will be combined to make multiple decisions, and we want these decisions to preserve the fairness properties of those underlying classifiers. Their work

introduced a number of models where fair mechanisms must be composed, demonstrated that naïve methods of composing fair algorithms do not preserve fairness, and identified strategies for preserving fairness in these settings. Importantly, these strategies treat the underlying classifiers as a *black-box* so that composition can be modular, eliminating the need to redesign the underlying classifier for every possible use.

Our work studies the archetypal *fair cohort selection problem* introduced by Dwork & Ilvento (2019). In this problem, we would like to select $k$ candidates for a job from a universe of $n$ possible candidates. We are given a classifier that assigns a *score* $s_i \in [0, 1]$ to each candidate,[1] with the guarantee that the scores are individually fair with respect to some similarity metric $\mathcal{D}$. We would like to select a set of $k$ candidates such that the probability $p_i$ of selecting any candidate should be fair with respect to the same metric.

**Preserving Fairness.** Since our goal is to treat the scores $s_1, \ldots, s_n$ as a black-box, without knowing the underlying fairness metric, we posit that the scores are fair with respect to some $\mathcal{D}$, i.e. they satisfy

$$\forall i, j, \ |s_i - s_j| \leq \mathcal{D}(i, j), \tag{1}$$

but that this metric is *unknown.* Thus, we will design our cohort selection algorithm so that its output probabilities $p_1, \ldots, p_n$ will satisfy

$$\forall i, j, \ |p_i - p_j| \leq |s_i - s_j| \leq \mathcal{D}(i, j) \tag{2}$$

as this guarantees that we preserve the underlying fairness constraint. Without further assumptions about the fairness metric, the only way to preserve fairness is to satisfy (2).

We can trivially solve the fair cohort selection problem by ignoring the scores and selecting a uniformly random set of $k$ candidates, so that every candidate is chosen with the same probability $k/n$. Thus, in this paper we propose to study algorithms for fair cohort selection that produce a cohort with optimal (or approximately optimal) utility subject to the fairness constraint. Specifically, we consider two variants of the problem:

**Linear Utilities.** For the linear utility model, we assume that

1. the utility for selecting a cohort is the sum of independently assigned utilities for each member, and

2. the scores given by the original classifier accurately reflect the utility of each member of the cohort.

Assumption 1 essentially says that members of the cohort are not complements or substitutes and assumption 2 makes sense if the task that we designed the classifier to solve and the task that the cohort is being used for are closely related. Under these assumptions, it is natural to define the utility for a cohort selection algorithm to be $\sum_i p_i s_i$, which is the expectation over the choice of the cohort of the sum of the scores across members in the cohort.

**Ratio Utilities.** For the ratio utility model, we continue to rely on assumption 1, but we now imagine that the expected utility for the cohort is some other linear function $\sum_i p_i u_i$ where the variables $u_i$ are some *unknown* measure of utility in $[0, 1]$. Assuming that for every candidate $i$ the score $s_i$ is generally correlated with the utility $u_i$ even if they are not exactly the same, we normalize by $\sum_i s_i u_i$, as the original scores can still be a reasonable choice of selection probabilities that satisfy the fairness constraint. Since the utility function is unknown, our goal is now to produce a cohort that maximizes the utility in the *worst case* over possible utility functions. As we show (Lemma 2.1), maximizing the worst-case utility amounts to assigning probabilities that maximize the quantity $\min_i p_i/s_i$. Optimizing the ratio utility implies a lower bound on the cohort utility under any utility function that satisfies assumption 1, including our linear utility above. However, it does not necessarily optimize the linear utility, thus the two utility models are distinct.

Our technical contributions are polynomial-time algorithms for fair and useful cohort selection with both linear and ratio utilities, in two different algorithmic models.

---

[1]In the model of Dwork et al. (2012), fairness requires that the classifier be randomized, so we can think of the output of the classifier as continuous scores rather than binary decisions.

**Offline Setting.** In this setting, we are given all the scores $s_1, \ldots, s_n$ at once, and must choose the optimal cohort. We present a polynomial-time algorithm for each model that computes a fair cohort with optimal expected utility.

**Online Setting.** In this setting, we are given the scores $s_1, s_2, \ldots$ as a stream. Ideally, after receiving $s_i$, we would like to immediately accept candidate $i$ to the cohort or reject them from the cohort. However, this goal is too strong (Dwork & Ilvento, 2019), so we consider a relaxation where candidate $i$ can be either rejected at any point during the online process, or kept on hold until later, and we would like to output a fair cohort at the end of the stream with optimal expected utility while minimizing the number of candidates who are kept on hold. We must keep at least $k$ candidates pending in order to fill our cohort. For ratio utilities we give a fair and optimal algorithm in this setting that keeps at most $O(k)$ candidates on hold. For linear utilities, we give an *approximately* fair and optimal algorithm. Specifically, we present a polynomial-time algorithm for this online setting where the fairness constraints are satisfied with an $\varepsilon$ additive error and utility is optimized up to an additive error of $O(k\sqrt{\varepsilon})$, for any desired $\varepsilon > 0$, while ensuring that the number of users on hold never exceeds $O(k + 1/\varepsilon)$. To interpret our additive error guarantee, since the fairness constraint applies to the probabilities $p_1, \ldots, p_n$, allowing the fairness constraint to be violated by an additive error of, say, $\varepsilon = 0.01$ means that every candidate is selected with probability that is within $\pm 1\%$ of the probability that candidate would have been selected by some fair mechanism.

## 1.1 Techniques

We start with a brief overview of the key steps in our algorithms for each setting. The formal description of the algorithms can be found in Sections 3 and 4. All omitted proofs are in the appendix.

**Offline Setting.** Our offline algorithm is based on two main properties of the fair cohort selection problem. First, we use a dependent-rounding procedure that takes a set of probabilities $p_1, \ldots, p_n$ such that $\sum_{i=1}^n p_i = k$ and outputs a random cohort $C$, represented by indicator random variables $\tilde{p}_1, \ldots, \tilde{p}_n \in \{0, 1\}$ with $\sum_{i=1}^n \tilde{p}_i = k$ such that $\mathbb{E}(\tilde{p}_u) = p_u$ for every candidate $u$. Thus, it is enough for our algorithm to come up with a set of marginal probabilities for each candidate to appear in the cohort, and then run this rounding process, rather than directly finding the cohort.

To find the marginal probabilities that maximize the linear utility with respect to the scores $s_1, \ldots, s_n \in [0, 1]$, we would like to solve the linear program (LP)

$$\text{maximize} \quad \sum_{i=1}^n p_i s_i$$

$$\text{s.t.} \quad \sum_{i=1}^n p_i \leq k$$
$$\forall i, j, |p_i - p_j| \leq |s_i - s_j|$$
$$\forall i, 0 \leq p_i \leq 1$$

In this LP, the first and third constraints ensure that the variables $p_u$ represent the marginal probability of selecting candidate $u$ in a cohort of size $k$. The second constraint ensures that these probabilities $p$ are fair with respect to the same measure $\mathcal{D}$ as the original scores $s$ (i.e. $|p_u - p_v| \leq |s_u - s_v| \leq \mathcal{D}(u, v)$). Although we could have used the stronger constraint $|p_u - p_v| \leq \mathcal{D}(u, v)$, writing the LP as we do means that our algorithm does not need to know the underlying metric, and that our solution will preserve any stronger fairness that the scores $s$ happen to satisfy.

While we could use a standard linear program solver, we can get a faster solution that is also useful for extending to the online setting by noting that this LP has a specific closed form solution based on "water-filling". Specifically, if $\sum_{i=1}^n s_i \leq k$, then the optimal solution simply adds some number $c \geq 0$ to all scores and sets $p_u = \min\{s_u + c, 1\}$, and an analogous solution works when $\sum_{i=1}^n s_i > k$.

For the ratio utility, although computing the optimal marginal probabilities does not correspond to solving a linear program, the solutions for the two utility functions are the same when $\sum_{i=1}^n s_i \leq k$. When $\sum_{i=1}^n s_i > k$, we maximize the ratio of marginal probability over score by scaling all the scores down by the same factor $k/\sum_{i=1}^n s_i$.

**Online Setting.** In the online setting we do not have all the scores in advance, thus for the linear utility we cannot solve the linear program, and do not even know the value of the constant $c$ that determines the solution. We give two algorithms for addressing this problem. The basic algorithm begins by introducing some *approximation*, in which we group users into $1/\varepsilon$ groups based on their scores, where group $g$ contains users with scores in $((g-1)\varepsilon, g\varepsilon]$. This grouping can only reduce utility by $O(k\sqrt{\varepsilon})$, and can only lead to violating the fairness constraint by $\varepsilon$. Since users in each bucket are treated identically, we know that when we reach the end of the algorithm, we can express the final cohort as containing a random set of $n_g$ members from each group $g$. Thus, to run the algorithm we use *reservoir sampling* to maintain a random set of at most $k$ candidates from each group, reject all other members, and then run the offline algorithm at the end to determine how many candidates to select from each group.

The drawback of this method is that it keeps as many as $k/\varepsilon$ candidates on hold until the end. Thus, we develop an improved algorithm that solves the linear program in an online fashion, and uses the information it obtains along the way to more carefully choose how many candidates to keep from each group. This final algorithm reduces the number of candidates on hold by as much as a quadratic factor, down to $2k + \frac{1}{\varepsilon}$.

For ratio utility, we are able to fully maximize the utility due to the way we decrease scores when the sum is greater than $k$. When the sum of scores is less than $k$, we must be careful in order to avoid being unfair to candidates eliminated early on. Thus, we maintain three groups; the top candidates, candidates undergoing comparisons, and candidates given uniform probability. The groups are dynamic and candidates can move from the first to the second and then the third group before getting rejected. Once the sum of scores exceeds $k$ the groups are no longer needed. The algorithm considers at most $O(k)$ candidates at any time.

## 1.2 Related Work

**Individual Fairness and Composition.** Our work fits into the line of work initiated by Dwork et al. (2012) on *individual fairness*, which imposes constraints on how algorithms may distinguish specific pairs of individuals. Within this framework, difficulties associated with composing fair algorithms were first explored by Dwork & Ilvento (2019), which introduced the fair-cohort-selection problem that we study. Issues of composition in pipelines were further studied by Dwork et al. (2020).

Because the scores figure into both our fairness constraint and utility measure, our work has some superficial similarity to work on *meritocratic fairness*. At a high-level meritocratic fairness (Joseph et al., 2016; Kearns et al., 2017; Joseph et al., 2017; Kleine Buening et al., 2022) is a kind of individual fairness where we assume there exists an intrinsic notion of merit and requires that candidates with higher merit be given no less reward than candidates with lower merit. In this work, we study a notion of individual metric fairness that, intuitively, tries to preserve the distances between the scores and not necessarily their order. Our algorithms turn out to also preserve the ordering, but meritocratically fair algorithms will not, in general, preserve the distances. We note that in our cases the scores $s_1, \ldots, s_n$ might or may not be reflective of true merit, since we assume that these scores satisfy individual fairness with respect to some underlying metric $\mathcal{D}$, which may or may not reflect meritocratic values. The work by Kleine Buening et al. (2022) on set selection under meritocratic fairness is the most related to our work. Yet, their setting is different as the authors make the assumption that the utility function is non-linear and it depends on the combination of the selected candidates. One of our main assumptions is that every individual contributes to the cohort utility separately. The differences between the two problems are also highlighted in their work.

**Individually Fair Ranking and Cohort Selection.** Our problem has some similarity to *fair ranking*, where we have to produce a permutation over all $n$ candidates instead of just selecting a small cohort of $k \ll n$ candidates. Any algorithm for ranking implies an algorithm for cohort selection, since we can select the top $k$ elements in the ranking as our cohort, and a cohort selected this way may inherit some notion of fairness from the ranking. However, we do not know of any ranking algorithms that can be used in this way to satisfy the notions of fairness and optimality we study in this work. In particular, the most closely related work on individually fair ranking by Bower et al. (2021) uses an incomparable formulation of the ranking problem, and it is not clear how to express our constraints in their framework. In addition to the difference in how fairness and utility are defined, we are not aware of any fair ranking papers that work in an online model similar to the one we study.

**Group Fairness.** A complementary line of work, initiated by Hardt et al. (2016) considers notions of *group fairness*, which imposes constraints on how algorithms may distinguish in aggregate between individuals from different groups. While our work is technically distinct from work on group fairness, cohort/set selection has also been studied from this point of view (Zehlike et al., 2017; Stoyanovich et al., 2018) and different approaches for fair ranking have been proposed (Yang & Stoyanovich, 2017; Celis et al., 2018; Singh & Joachims, 2018; Yang et al., 2019; Zehlike & Castillo, 2020). Issues of composition also arise in this setting, as noted by several works (Bower et al., 2017; Kannan et al., 2019; Arunachaleswaran et al., 2021).

## 2 Preliminaries

### 2.1 Models

We consider the problem of selecting a cohort using the output of an individually fair classifier as defined in Dwork & Ilvento (2019).

**Definition 2.1** (Individual Fairness (Dwork et al., 2012)). Given a universe of individuals $U$ and a metric $\mathcal{D}$, a randomized binary classifier $C : U \to \{0, 1\}$ is individually fair if and only if for all $u, v \in U$, $|\mathbb{P}(C(u) = 1) - \mathbb{P}(C(v) = 1)| \leq \mathcal{D}(u, v)$.

We restate the formal definition of the problem for convenience.

**Definition 2.2** (Fair Cohort Selection Problem). Given a universe of candidates $U$, an integer $k$ and a metric $\mathcal{D}$, select a set $S$ of $k$ individuals such that the selection is individually fair with respect to $\mathcal{D}$, i.e. for all pairs of candidates $u, v \in U$, $|\mathbb{P}(u \in S) - \mathbb{P}(v \in S)| \leq \mathcal{D}(u, v)$.

In this work, we are interested in the setting of fairness under composition, and designing algorithms that exploit the fairness properties of their components instead of having direct access to $\mathcal{D}$. An algorithm which solves this problem will either preserve or shrink the pairwise distances between the selection probabilities of candidates. We can think of the probability of $C$ selecting a candidate $u$ as a continuous score $s_u$.

**Definition 2.3** (Fairness-Preserving Cohort Selection Problem). Given a classifier $\mathcal{C}$ who assigns score $s_i$ to candidate $i$, select a cohort of $k$ individuals where each individual $i$ has marginal selection probability $p_i$ such that for any pair of individuals $i, j$, $|p_i - p_j| \leq |s_i - s_j|$.

In some contexts that we define below, we consider a relaxation of this definition by adding a small error $\varepsilon$ to the pairwise distances.

**Definition 2.4** ($\varepsilon$-Approximate Fairness-Preserving Cohort Selection Problem). Given a classifier $\mathcal{C}$ who assigns score $s_i$ to any individual $i$ and a parameter $\varepsilon \in (0, 1]$, select a cohort of $k$ individuals where each individual $i$ has marginal selection probability $p_i$ such that for all pairs of individuals $i, j$ $|p_i - p_j| \leq |s_i - s_j| + \varepsilon$.

When the initial classifier $C$ is individually fair and the cohort selection algorithm preserves fairness $\varepsilon$-approximately, then the cohort selection probabilities satisfy $\varepsilon$-individual fairness.

**Definition 2.5** ($\varepsilon$-Individual Fairness). Given a universe of individuals $U$, a metric $\mathcal{D}$ and an $\varepsilon$ in $[0, 1]$, a randomized binary classifier $C : U \to \{0, 1\}$ is $\varepsilon$-individually fair if and only if for all $u, v \in U$, $|\mathbb{P}(C(u) = 1) - \mathbb{P}(C(v) = 1)| \leq \mathcal{D}(u, v) + \varepsilon$.

The problem of fairness-preserving cohort selection may be studied in an *offline* setting, where the universe of candidates is known in advance, as well as in an *online* setting, in which candidates arrive one-at-a-time. Dwork & Ilvento (2019) prove that making immediate selection decisions while preserving fairness is impossible, so we relax this to a setting in which we do not need to give immediate decisions as candidates arrive. To control the degree of relaxation, we consider the number of candidates who wait for a response during the selection stage; we seek to minimize the number of these pending candidates.

Individual fairness can be achieved by making uniform random selections among candidates, which is trivially possible in the offline case, and can be implemented in the online setting using reservoir sampling (Vitter, 1985). Therefore, it is important to consider an extension to the cohort selection problem in which we also want to maximize the utility of the selected cohort. We consider two measures of utility and construct

algorithms for cohort selection which optimize these measures. Let the selection scores for individuals under classifier $C$ be $s_1, \ldots, s_n$, and let their marginal probabilities of selection under an algorithm $A$ be $p_1, \ldots, p_n$.

**Utilities Correlated with Classifier Output.** We may consider that the classifier scores $s$ directly indicate the qualifications of an individual and that the utility of a cohort is the sum of the scores of each member. Then, our evaluation measure becomes the expected utility of the selected cohort.

**Definition 2.6** (Linear Utility). Consider a set of $n$ candidates whose scores from classifier $C$ are $s_1, \ldots, s_n$ and whose selection probabilities by algorithm $A$ are $p_1, \ldots, p_n$. We define the linear utility of $A$ to be $\text{Utility}_{\text{LINEAR}}(A, \{s_i\}) = \sum_{i=1}^{n} s_i p_i$.

As seen in Example 2.1, the maximum linear utility under the fairness-preserving constraint is not a monotone function of the input scores; as the score $s_i$ increases, the maximum linear utility may either increase, decrease, or stay constant. Yet, we show in Lemma 3.1 that it is robust to small input perturbations. Therefore, it can handle minor noise or errors in the input value. We exploit this property to develop an $\varepsilon$-individually fair algorithm for online cohort selection.

**Example 2.1.** Suppose we want to select 2 people from a set of 4 candidates and classifier $C$ assigns scores $s_1 = 0.1, s_2 = 0.3, s_3 = 0.6$ and $s_4 = 0.9$. The optimal fairness-preserving solution has marginal selection probabilities $p_1 = 0.125, p_2 = 0.325, p_3 = 0.625$ and $p_4 = 0.925$ and utility $\sum_{i=1}^{4} p_i s_i = 1.3175$. By increasing the score of the first individual to $s_1' = 0.3$, the cohort selection probabilities become $p_1' = 0.275, p_2' = 0.275, p_3' = 0.575$ and $p_4' = 0.875$. The new utility is $\sum_{i=1}^{4} p_i' s_i' = 1.2975$, which is lower than the previous one.

**Arbitrary Utilities.** Alternatively, we may consider the case where there exists an arbitrary, unknown utility value for each candidate $u_1, \ldots, u_n$ in $[0, 1]$, which may differ from the scores $s$. Given $s_i$ and $u_i$, a simple quantity to study is the ratio utility achieved by an algorithm $A$: $\frac{\sum_i p_i u_i}{\sum_i s_i u_i}$. Since we do not know the distribution of utility, we must consider the utility achieved by an algorithm $A$ for a worst-case utility vector $\vec{u}$.

**Lemma 2.1.** *The worst case ratio utility is equal to the minimum ratio of selection probability and classifier score of one candidate* $\min_{\vec{u}} \frac{\sum_i p_i u_i}{\sum_i s_i u_i} = \min_i \frac{p_i}{s_i}$.

We use this equivalence to redefine the ratio utility as follows.

**Definition 2.7** (Ratio Utility). Given $n$ candidates with scores $s_i$ and a cohort selection algorithm $A$ with marginal selection probabilities $p_i$, let the ratio utility of an algorithm $A$ be defined according to the coordinate with the minimum ratio: $\text{Utility}_{\text{RATIO}}(A, \{s_i\}) = \min_i \frac{p_i}{s_i}$.

Of course other utility measures exist; we study the measures defined above because they capture a natural intuition about real world scenarios. We present an example to demonstrate the properties of these two utility functions and the performance of pre-existing cohort selection algorithms.

**Example 2.2.** Consider the case where we would like to select 2 out of 3 candidates with scores $s_1 = 0.5$, $s_2 = 0.5$, $s_3 = 1$. The optimal solution in terms of ratio and linear utility is to pick $\{1, 3\}$ and $\{2, 3\}$, each with probability 0.5. This solution achieves ratio utility 1 and linear utility 1.5. The weighted sampling algorithm by Dwork & Ilvento (2019) selects each subset with probability proportional to its weight, and would select $\{1, 2\}$ with probability $1/4$, $\{1, 3\}$ with probability $3/8$, and $\{2, 3\}$ with probability $3/8$. The ratio utility of this solution is $3/4$ and its linear utility is $11/8$. PermuteThenClassify by Dwork & Ilvento (2019) selects $\{1, 2\}$ with probability $1/12$ and $\{1, 3\}$ or $\{2, 3\}$ with probability $11/24$ each. This solution achieves ratio utility $11/12$ and linear utility $35/24$. Hence, we see that weighted sampling and PermuteThenClassify are sub-optimal for both utility definitions.

## 2.2 Dependent Rounding

Our cohort selection algorithms for ratio and linear utility in the offline and online settings are based on a simple dependent rounding algorithm described in the appendix. This algorithm makes one pass over a list

of numbers in $[0, m]$, where $m \in \mathbb{R}$, and rounds them so that their sum is preserved and the expected value of each element after rounding is equal to its original value. The algorithm operates on two entries at a time - call these $a$ and $b$. If $a + b \le m$, the algorithm randomly rounds one of them to $a + b$ and the other one to 0 with the appropriate probabilities. If $a + b > m$, it rounds one to $m$ and the other to the remainder.

This rounding procedure is a special case of rounding a fractional solution in a matroid polytope (in this case, we have a uniform matroid). This problem has been studied extensively with rounding procedures satisfying additional desirable properties (see e.g. Chekuri et al. (2010)). Here we use a simple and very efficient rounding algorithm for the special case of the problem arising in our work.

**Lemma 2.2.** *Let $a_1, \ldots, a_n$ be a list of numbers in $[0, m]$ with $x = \sum_{i=1}^{n} a_i$. There is a randomized algorithm that outputs $\tilde{a}_1, \ldots, \tilde{a}_n \in [0, m]$ such that $\lfloor \frac{x}{m} \rfloor$ of the $a_i$ will be equal to $m$, one $a_i$ will be $x - \lfloor \frac{x}{m} \rfloor m$, the rest will be equal to 0, and for all $i \in \{1, \ldots, n\}$, $\mathbb{E}[\tilde{a}_i] = a_i$. When $m = 1$, we call this algorithm* `round-pr` *and when $a_1, \ldots, a_n$ are natural numbers we call it* `round-nat`.

We use the rounding procedure in two ways. We call `round-pr` when we want to round a list of real numbers in $[0, 1]$ which represent selection probabilities and `round-nat` to round natural numbers which correspond to counts.

**Corollary 2.1.** *If $m = 1$ and $\sum_{i=1}^{n} a_i = k \in \mathbb{N}$, then after the dependent rounding of* `round-pr` *$k$ elements will be equal to 1 and the rest will be equal to 0.*

# 3 Linear Utility

The first definition of utility we study is that of linear utility. In this setting, the aim of the cohort selection is to maximize the expected sum of scores of the group of $k$ people selected while preserving fairness. We can formulate this problem as a linear program, as shown in Section 1.1.

The key idea of the solution is to move all the selection scores up or down by the same amount, depending on the value of their sum, and clip the probabilities that are less than zero or over one. The algorithm we propose for the online setting solves approximately a relaxation of the cohort selection problem, where the probabilities of two candidates can differ by at most the given metric plus a small constant $\varepsilon$. The new problem is defined as the linear program presented in Section 1.1, where the second constraint, $\forall i, j \in [n], |p_i - p_j| \le |s_i - s_j|$, is replaced by $\forall i, j \in [n], |p_i - p_j| \le |s_i - s_j| + \varepsilon$.

## 3.1 Offline Cohort Selection

In the offline setting, the algorithm has full access to the set of candidates and their scores from classifier $C$. By Corollary 2.1, if the classifier's scores sum up to $k$, then the rounding algorithm can form a cohort by simply using the input scores as the marginal probabilities of selection. Yet, the sum of the scores assigned by $C$ might not be $k$. If it is less than $k$, Algorithm 1 moves all the probabilities up by the same amount and clips those that exceed 1, so that their sum becomes equal to $k$. In more detail, the marginal selection probability of candidate $i$ becomes $p_i = \min\{s_i + c, 1\}$, where constant $c$ is such that $\sum_{i=1}^{n} p_i = k$. This adjustment preserves the probability differences between the pairs of candidates, unless some candidate gets assigned probability 1, in which case some differences decrease. As a result, the differences of the new probabilities will be bounded by the same metric as the scores. The case where the sum of scores is greater than $k$ is treated similarly. More specifically, the new marginal probabilities are of the form $p_i = \max\{0, s_i - c\}$. After the adjustment, Algorithm 1 selects the cohort by running the dependent rounding procedure.

**Theorem 3.1.** *For any classifier $C$ that assigns scores $s_1, \ldots, s_n$ to $n$ candidates, Algorithm 1 solves the fairness-preserving cohort selection problem by selecting $k$ candidates with marginal probabilities $p_1, \ldots, p_n$ which achieve optimal linear utility $\sum_{i=1}^{n} p_i s_i$.*

**Corollary 3.1.** *If $C$ is individually fair, then Algorithm 1 is individually fair.*

## 3.2 Online Cohort Selection

For the online setting of the problem with linear utility, we present Algorithm 2 which reads $C$'s output from a stream and selects a cohort with high utility while deferring the decision for a small number of candidates.

---

**Alg. 1:** Offline Cohort Selection for Linear Utility

---

**Input:** scores $s_1, \ldots, s_n \in [0,1]$ from $C$, cohort size $k$

**Output:** set of accepted candidates

1   sum $\leftarrow \sum_{i=1}^n s_i$

2   **if** sum $< k$ **then**

    // move scores up

3     $c \leftarrow \frac{k - \text{sum}}{n}$; $p_i \leftarrow s_i + c$, $\forall i \in [n]$

4     **while** $\exists p_i > 1$ **do**

5       $\mathcal{S}_{<1} \leftarrow \{j : p_j < 1\}$

6       $p_j \leftarrow p_j + \frac{p_i - 1}{|\mathcal{S}_{<1}|}$, $\forall j \in \mathcal{S}_{<1}$

7       $p_i \leftarrow 1$

8   **else**

    // move scores down

9     $c \leftarrow \frac{\text{sum} - k}{n}$; $p_i \leftarrow s_i - c$, $\forall i \in [n]$

10    **while** $\exists p_i < 0$ **do**

11      $\mathcal{S}_{>0} \leftarrow \{j : p_j > 0\}$

12      $p_j \leftarrow p_j + \frac{p_i}{|\mathcal{S}_{>0}|}$, $\forall j \in \mathcal{S}_{>0}$

13      $p_i \leftarrow 0$

    // Now list sums to $k$ and all $p_i \in [0,1]$

14   $\{\tilde{p}_i\} \leftarrow$ round-pr$(p_1, \ldots, p_n, 1)$ // select cohort of size $k$

15   **return** candidates with non-zero $\tilde{p}_i$

---

While processing the stream, this algorithm only rejects candidates and releases all the acceptances at the end of the stream. The main difference in comparison to the offline cohort selection algorithm is that Algorithm 2 splits the candidates into groups with similar scores, within an $\varepsilon$ interval, and treats any member of a group as equivalent. Hence, the final probability of being selected in the cohort is equal for any candidate of the same group. This leads to a relaxation of the problem, which we call $\varepsilon$-approximate fairness-preserving cohort selection problem.

As we mentioned, the algorithm splits the candidates from the stream into groups according to their score from $C$. Particularly, the interval $(0,1]$ is divided into $m$ intervals of size $\varepsilon$ each, where $m = \lceil \frac{1}{\varepsilon} \rceil$. The $i$-th candidate gets assigned to group 0 if $s_i = 0$ or group $g \in \{1, \ldots, m\}$ if $s_i \in ((g-1)\varepsilon, g\varepsilon]$. Once candidate $i$ is in the group, their score gets rounded up and their new score $\hat{s}_i$ is the same as that of all the other members of the group, $\hat{s}^g = g\varepsilon$. The use of the groups affects the performance of the algorithm in terms of not only individual fairness, but also utility. Lemma 3.1 shows that the decrease in utility is caused only by the use of rounding, as Algorithm 2 achieves the same linear utility as the corresponding offline algorithm when the scores are rounded to multiples of $\varepsilon$.

**Lemma 3.1.** *Given selection scores $s_1, \ldots, s_n$ from $C$, for a set of $n$ candidates and a parameter $\varepsilon \in (0,1]$, we split $(0,1]$ into $m = \lceil \frac{1}{\varepsilon} \rceil$ intervals of length $\varepsilon$ and for all $i \in [n]$ we set $\hat{s}_i = g\varepsilon$, where $g$ is such that $s_i \in ((g-1)\varepsilon, g\varepsilon]$. When Algorithm 1 runs for input $\hat{s}_1, \ldots, \hat{s}_n$ and cohort size $k$ it solves the $\varepsilon$-approximate fairness-preserving cohort selection problem with marginal selection probabilities $p_1, \ldots, p_n$, and it achieves linear utility $\sum_{i=1}^n p_i s_i \geq \sum_{i=1}^n p_i^* s_i - k(\varepsilon + 2\sqrt{\varepsilon})$, where $p_1^*, \ldots, p_n^*$ is the optimal solution for the offline fairness-preserving problem for input $s_1, \ldots, s_n$.*

Since for each group we keep only some people on hold, we preserve the probability information of the rejected candidates by considering that each pending candidate represents themselves and a fraction of the rejected people from their group. The fraction of people represented by the $i$-th candidate is denoted by $n_i$. One approach is to maintain at most $k$ people pending per group uniformly at random and have every candidate within a certain group represent the same fraction of people. This basic algorithm keeps $k/\varepsilon$ candidates on hold. Our algorithm reduces the number of people pending by allowing candidates of the

---

**Alg. 2:** Online Cohort Selection for Linear Utility

---

**Input:** stream of scores $s_1, \ldots, s_n \in [0,1]$ from $C$, cohort size $k$, constant $\varepsilon \in (0,1]$
**Output:** set of accepted candidates

1  $n \leftarrow 0$; sum $\leftarrow 0$// number of candidates and sum of scores
2  $n^g \leftarrow 0$, for all groups $g \in [[\frac{1}{\varepsilon}]]$// size of group $g$
3  **for** *candidate $i$ in stream* **do**
4  $\quad$ add $i$ to group $g$ where $s_i \in ((g-1)\varepsilon, g\varepsilon]$
5  $\quad$ $n^g \leftarrow n^g + 1$
6  $\quad$ $\hat{s}_i \leftarrow g\varepsilon$; sum $\leftarrow$ sum $+ \hat{s}_i$
7  $\quad$ $n_i \leftarrow 1$; $n \leftarrow n+1$
$\quad$ // Compute group scores for this input
8  $\quad$ **if** sum $< k$ **then**
9  $\quad\quad$ $c \leftarrow \frac{k-sum}{n}$
10 $\quad\quad$ $s^g \leftarrow g\varepsilon + c$, for all groups $g$
11 $\quad\quad$ **while** $\exists$ *group $g$ with $s^g > 1$* **do**
12 $\quad\quad\quad$ $\mathcal{F}_{<1} \leftarrow \{\text{group } f : s^f < 1\}$
13 $\quad\quad\quad$ $n_{<1} \leftarrow \sum_{f \in \mathcal{F}_{<1}} n^f$
14 $\quad\quad\quad$ $s^f \leftarrow s^f + n^g \frac{(s^g - 1)}{n_{<1}}, \forall f \in \mathcal{F}_{<1}$
15 $\quad\quad\quad$ $s^g \leftarrow 1$
16 $\quad$ **else if** sum $> k$ **then**
17 $\quad\quad$ $c \leftarrow \frac{sum-k}{n}$
18 $\quad\quad$ $s^g \leftarrow g\varepsilon - c$, for all groups $g$
19 $\quad\quad$ **while** $\exists$ *group $g$ with $s^g < 0$* **do**
20 $\quad\quad\quad$ $\mathcal{F}_{>0} \leftarrow \{\text{group } f : s^f > 0\}$
21 $\quad\quad\quad$ $n_{>0} \leftarrow \sum_{f \in \mathcal{F}_{>0}} n^f$
22 $\quad\quad\quad$ $s^f \leftarrow s^f + n^g \frac{s^g}{n_{>0}}, \forall f \in \mathcal{F}_{>0}$
23 $\quad\quad\quad$ $s^g \leftarrow 0$
24 $\quad$ **else** $s^g \leftarrow g\varepsilon$, for all groups $g$
25 $\quad$ **for** *group $g$* **do**
26 $\quad\quad$ **if** $s^g > 0$ **then**
27 $\quad\quad\quad$ $\{n_i : i \text{ in } g\} \leftarrow$ round-nat($\{n_i : i \text{ in g}\}, v = \lfloor \frac{1}{s^g} \rfloor$)
$\quad\quad\quad$ // we ensure $n_i v \leq 1$
28 $\quad\quad\quad$ reject candidate $i$ if $n_i = 0$
29 $\quad\quad$ **else**
30 $\quad\quad\quad$ reject all candidates in $g$

31 $\tilde{s}_i \leftarrow n_i s^g$, for all groups $g$ and candidates $i$ in $g$
$\quad$ // Now candidate scores sum up to $k$ and all $\tilde{s}_i$ in $[0,1]$
32 $\{\tilde{s}_i\} \leftarrow$ round-pr($\{\tilde{s}_i : i \text{ in any group}\}, 1$)// select cohort of size $k$
33 **return** candidates with non-zero $\tilde{s}_i$

---

same group to represent varying fractions of people. Procedure round-nat is used to eliminate candidates and determine what fraction of people's selection probabilities each candidate represents.

**Theorem 3.2.** *For any classifier $C$ that assigns scores $s_1, \ldots, s_n$ to $n$ candidates, Algorithm 2 solves the online $\varepsilon$-approximate fairness-preserving cohort selection problem for any $\varepsilon \in (0,1]$ by selecting a cohort of size $k$ with marginal probabilities $p_1, \ldots, p_n$, achieves linear utility $\sum_{i=1}^n p_i s_i \geq \sum_{i=1}^n p_i^* s_i - k(\varepsilon + 2\sqrt{\varepsilon})$ (where $p_1^*, \ldots, p_n^*$ is the optimal solution for the offline fairness-preserving problem for input $s_1, \ldots, s_n$), and keeps at most $O(k + \frac{1}{\varepsilon})$ candidates pending.*

**Corollary 3.2.** *If $C$ is individually fair, then Algorithm 2 is $\varepsilon$-individually fair.*

## 4 Ratio Utility

In Definition 2.7 we consider the ratio utility obtained by a selection algorithm, and show that this is controlled by the minimum ratio of $p_i$ and $s_i$.

### 4.1 Offline Cohort Selection

We begin with the scenario of selecting $k$ of $n$ individuals when all candidates are known in advance. Dwork & Ilvento (2019) give a solution to this scenario (with non-optimal utility, as shown in Example 2.2); we present Algorithm 3, an alternate one that achieves the optimal ratio utility. Notice that the sum of classifier scores $\sum_i s_i$ need not add up to $k$. If the sum exceeds $k$, the algorithm simply scales down all probabilities so that the new sum is $k$. It is not hard to show that this operation preserves fairness and gives optimal utility. This multiplicative scaling will not work when the sum is smaller than $k$, however, since it increases the probability difference among candidates and can be unfair. Intuitively, the solution in this case is to additively increase all probabilities by the same amount, thus preserving fairness. However, some care is needed, as no probability can exceed 1.

---

**Alg. 3:** Offline Cohort Selection for Ratio Utility

**Input:** scores $s_1, \ldots, s_n \in [0,1]$ from $C$, cohort size $k$
**Output:** set of accepted candidates
1  sum $\leftarrow \sum_{i=1}^{n} s_i$
2  **if** *sum $< k$* **then**
    // move scores up
3      $c \leftarrow \frac{k-\text{sum}}{n}; \; p_i \leftarrow s_i + c, \forall i \in [n]$
4      **while** $\exists p_i > 1$ **do**
5          $\mathcal{S}_{<1} \leftarrow \{j : p_j < 1\}$
6          $p_j \leftarrow p_j + \frac{p_i - 1}{|\mathcal{S}_{<1}|}, \forall j : p_j < 1$
7          $p_i \leftarrow 1$
8  **else**
    // move scores down
9      $p_i \leftarrow s_i \cdot \frac{k}{\text{sum}}, \forall i \in [n]$
    // Now list sums to $k$ and all $p_i \in [0,1]$
10  $\{\tilde{p}_i\} \leftarrow$ round-pr$(p_1, \ldots, p_n, 1)$ // select cohort of size $k$
11  **return** candidates with non-zero $\tilde{p}_i$

---

**Lemma 4.1.** *If sum $< k$, Algorithm 3 increases each input score $s_i$ to a final output $p_i = s_i + \alpha_i \geq s_i$ such that all of the candidates $j$ with $p_j < 1$ receive the same cumulative adjustment value $\alpha_j = v$.*

**Theorem 4.1.** *For any classifier $C$ that assigns scores $s_1, \ldots, s_n$ to $n$ candidates, Algorithm 3 solves the fairness-preserving cohort selection problem by selecting $k$ candidates with marginal probabilities $p_1, \ldots, p_n$ that achieve the optimal value of ratio utility $\min_i \frac{p_i}{s_i}$.*

**Corollary 4.1.** *If $C$ is individually fair, then Algorithm 3 is individually fair.*

### 4.2 Online Cohort Selection

We now consider the online scenario, in which our goal is to keep as few candidates pending as possible. It is clear that the number of candidates on hold must be at least $k$ since we need to accept $k$ candidates at the end. We provide an algorithm that achieves optimal ratio utility for the online fairness-preserving cohort selection problem, while leaving only $O(k)$ candidates pending. We use a parameter $\alpha \in [0, 1/2]$, which controls the number of candidates pending, with $\alpha = 1/2$ by default. The update procedure depends on the sum of scores of the candidates seen so far. We maintain a set of $k/\alpha$ candidates with highest scores called TOP, and two sets REST and RAND of the remaining candidates. A new candidate $i$ is added to either TOP (and thus bumps some other candidate from TOP to REST), or they are added to REST. We make sure

---

**Alg. 4:** Online Cohort Selection for Ratio Utility

---

**Input:** stream of scores $s_1, \ldots, s_n \in [0, 1]$ from $C$, cohort size $k$, constant $\alpha \in [0, 1/2]$
**Output:** set of accepted candidates

**1** sum $\leftarrow 0$
**2** TOP $\leftarrow \emptyset$; REST $\leftarrow \emptyset$; RAND $\leftarrow \emptyset$
**3** **for** *candidate $i$ in stream* **do**
**4**   sum $\leftarrow$ sum $+ s_i$;   $p_i \leftarrow s_i$
**5**   **if** *sum $< k$* **then**
**6**    **if** *$|TOP| < \lceil k/\alpha \rceil$* **then** add $i$ to TOP
**7**    **else if** *$min(s_j)$ in TOP $< s_i$* **then**
**8**     move min element from TOP to REST
**9**     add $i$ to TOP
**10**    **else** add $i$ to REST
**11**    $\{p_i : i \in \text{REST}\} \leftarrow$ round-pr$(\{p_i : i \in \text{REST}\}, 1 - \alpha)$
**12**    **for** *$j$ in REST with $p_j = 0$* **do**
**13**     remove $j$ from REST
**14**     add $j$ to unif. reservoir sample of size $k$
**15**     reject candidate that was eliminated from the unif. reservoir sample
**16**    set RAND to unif. reservoir sample
**17**   **else**
**18**    **if** *first occurrence of sum $\geq k$* **then**
**19**     reject all candidates in RAND and remove RAND
**20**     store TOP and REST in PENDING
**21**     incr $\leftarrow \frac{k}{\text{sum}}$ ;   scale $\leftarrow \frac{k}{\text{sum}}$
**22**    **else**
**23**     incr $\leftarrow \frac{\text{sum} - s_i}{\text{sum}}$;   scale $\leftarrow$ scale $\cdot$ incr
**24**    $p_i \leftarrow p_i \cdot$ scale
**25**    $p_j \leftarrow p_j \cdot$ incr, $\forall j \in$ PENDING
**26**    add $i$ to PENDING
**27**    $\{p_i : i \in \text{PENDING}\} \leftarrow$ round-pr$(\{p_i : i \in \text{PENDING}\}, 1)$
**28**    reject $j$ and remove it from PENDING if $p_j = 0$

**29** **if** *sum $< k$* **then**
**30**   $c \leftarrow \frac{k - \text{sum}}{n}$;   $p_i \leftarrow p_i + c, \forall i \in$ TOP $\cup$ REST
**31**   $p_i \leftarrow \frac{k - \sum_{i \in \text{TOP} \cup \text{REST}} p_i}{|\text{RAND}|}, \forall i \in$ RAND
**32**   **while** *$\exists p_i > 1$* **do**
**33**    $\mathcal{S}_{(\geq 1)} \leftarrow \{i : p_i \geq 1\}$; $c \leftarrow \frac{p_i - 1}{n - |\mathcal{S}_{(\geq 1)}|}$
**34**    $p_j \leftarrow p_j + c, \forall j \in$ TOP $\cup$ REST with $p_j < 1$
**35**    set $n_{\text{modified}}$ to number of modified people
**36**    $p_j \leftarrow p_j + \frac{c \cdot (n - |\mathcal{S}_{(\geq 1)}| - n_{\text{modified}})}{|\text{RAND}|}, \forall j \in$ RAND
**37**    $p_i \leftarrow 1$
**38**   $\{p_i : i \in \text{TOP} \cup \text{REST} \cup \text{RAND}\} \leftarrow$ round-pr$(\{p_i : i \in \text{TOP} \cup \text{REST} \cup \text{RAND}\}, 1)$
**39**   set PENDING to indices of nonzero $p_i$

**40** **return** PENDING

---

the list REST is not too big by rounding and eliminating candidates. Eliminated candidates are sampled uniformly to enter RAND, which has size $k$.

During the online phase, it is tempting to use the idea from the offline case to eliminate candidates; take two candidates from REST with scores $a$ and $b$ and round them to $a + b$ and 0. However, we must be careful

when rounding REST to ensure that probabilities will not later exceed 1 if the stream ends with sum $< k$. The key idea (and motivation for having TOP) is that candidates not in TOP have increments at most $\alpha$, which we can see as follows. For REST to be non-empty, we must have at least $n > k/\alpha$ candidates. Let $x$ be the number of candidates who are in TOP when the algorithm finishes and achieve probability 1. For each of the other candidates, the increment $p_i - s_i$ is less than $(k - x)/(n - x) < k/n$, since $k < n$. Thus that cumulative increment is at most $\alpha$, and it is safe to round probabilities in REST to 0 and $1 - \alpha$.

If the stream ends with $\sum_i s_i < k$, we will increase the scores of everyone pending, and use RAND to compensate for already eliminated candidates. Let $A$ be the set of candidates in TOP and REST and $\bar{A}$ be the set of all others seen. As in the offline case, we will evenly increase the probabilities of all candidates by adding the appropriate $c$ to each candidate, and using water-filling where some candidates reach 1. For candidates in $A$, this proceeds exactly as in the offline case. For candidates in $\bar{A}$, we increase the probabilities without explicitly storing all of them by maintaining a set called RAND consisting of $k$ uniformly random candidates. Each member of RAND receives probability equal to $1/k$ times the sum of the probabilities of the candidates in $\bar{A}$. Since everyone in $\bar{A}$ has the same probability, this rounding clearly preserves the marginal probabilities. Finally, we use the offline algorithm to select the cohort from the union of RAND and $A$.

The case where the sum of probabilities exceeds $k$ is simpler. The algorithm always scales down all probabilities so that they add up to exactly $k$. When a new candidate appears, they receive a probability equal to what $C$ gives them, times the current scaling factor, and get added to TOP $\cup$ REST. The algorithm then scales down all probabilities so that the sum is exactly $k$ and reduces the size of TOP $\cup$ REST by repeatedly rounding pairs of candidate as in the offline setting. In this case, the set RAND is not used.

**Theorem 4.2.** *For any classifier $C$ that assigns scores $s_1, \ldots, s_n$ to $n$ candidates, Algorithm 4 solves the fairness-preserving cohort selection problem by selecting $k$ candidates with marginal probabilities $p_1, \ldots, p_n$ that achieve the optimal value of ratio utility. The algorithm leaves no more than $O(k)$ candidates pending at any time.*

**Corollary 4.2.** *If $C$ is individually fair, then Algorithm 4 is individually fair.*

## 5 Conclusion

In this work, we defined two notions of utility for the problem of cohort selection, linear and ratio utility, and designed polynomial time algorithms that optimize them (or approximately optimize them) while preserving the fairness of the given classifier. This means that if the given classifier is individually fair, then our cohort selection algorithms are also individually fair with respect to the same metric. We studied two settings, the standard offline setting and an online setting where we can reject candidates at any stage of the process and accept candidates only at the end, keeping the number of candidates that are waiting for the decision low. While our algorithms for ratio utility and offline linear utility are optimal in terms of fairness and utility, our online algorithm for linear utility achieves $\varepsilon$-individual fairness with an additive error in the utility and the number of candidates kept on hold.

Overall, our approach is based on the assumptions that every candidate contributes to the utility of the cohort separately and that the candidate utilities are correlated with their selection scores given by the individually fair classifier. As the linear utility requires the assumption that the utilities of the candidates are their classification scores, we tried to relax it by defining the ratio utility in which the utilities are unknown but still correlated with the classification scores. The natural extension of this work would be to design algorithms that optimize a utility function where the candidate utilities are known and decoupled from the classification scores. Another interesting direction would be to relax our first assumption and study utility functions where the candidates do not contribute separately, but the cohort utility depends on the combination of candidates selected.

## Acknowledgments

KB and JU were supported by NSF awards CCF-1750640 and CNS-2120603. KB and HN were supported by NSF grant CCF-1750716.

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

## A    Proofs and Algorithm from Section 2

### A.1    Proof of Lemma 2.1

**Lemma 2.1.** The worst case ratio utility is equal to the minimum ratio of selection probability and classifier score of one candidate $\min_{\vec{u}} \frac{\sum_i p_i u_i}{\sum_i s_i u_i} = \min_i \frac{p_i}{s_i}$.

*Proof.* Let $j$ be the index of the candidate with the smallest ratio of model output and classifier score, i.e. $\frac{p_j}{s_j} \leq \frac{p_i}{s_i}, \forall i \neq j$. Compare the minimum over all utility vectors to a particular choice of utility vector $u^*$ satisfying $u_j^* = 1$, and $u_{i \neq j}^* = 0$. We know that the ratio for any particular vector will be at least as large as the minimum: $\min_{\vec{u}} \frac{\sum_i p_i u_i}{\sum_i s_i u_i} \leq \frac{\sum_i p_i u_i^*}{\sum_i s_i u_i^*} = \frac{p_j u_j^*}{s_j u_j^*} = \frac{p_j}{s_j} = \min_i \frac{p_i}{s_i}$.

We can also obtain a bound from the other direction. Let $m$ be the smallest ratio between model output and classifier score $m = \min_i p_i/s_i$ and $u \in [0,1]^n$ be any utility vector with $\sum_i u_i > 0$. Then, for all $i \in [n]$ we have $p_i u_i \geq m s_i u_i$. Therefore, $min_{\vec{u}} \frac{\sum_i p_i u_i}{\sum_i s_i u_i} \geq \min_i \frac{p_i}{s_i}$    □

### A.2    Dependent Rounding Algorithm

Here we include a detailed description of the dependent rounding algorithm (Algorithm 5) mentioned in Section 2.2.

### A.3    Proof of Lemma 2.2

**Lemma 2.2.** Let $a_1, \ldots, a_n$ be a list of numbers in $[0, m]$ with $x = \sum_{i=1}^n a_i$. There is a randomized algorithm that outputs $\tilde{a}_1, \ldots, \tilde{a}_n \in [0, m]$ such that $\lfloor \frac{x}{m} \rfloor$ of the $a_i$ will be equal to $m$, one $a_i$ will be $x - \lfloor \frac{x}{m} \rfloor m$, the

---

**Algorithm 5:** Rounding

---

**Input:** a list of $n$ numbers $a_1, a_2, \ldots, a_n \in \mathbb{R}_{\geq 0}$ and the maximum value of a rounded number $m \in \mathbb{R}_{\geq 0}$
**Output:** list of rounded numbers $\tilde{a}_1, \tilde{a}_2, \ldots, \tilde{a}_n \in \mathbb{R}_{\geq 0}$

---

**1** pendingIndex $\leftarrow 1$
**2 for** $i$ *from* $2$ *to* $n$ **do**
**3**     **if** $a_i = 0$ *and* $a_{\text{pendingIndex}} = 0$ **then** continue to next $i$
**4**     $a \leftarrow a_i$ ; $b \leftarrow a_{\text{pendingIndex}}$
**5**     choose $u$ randomly from $Unif(0,1)$
**6**     **if** $a + b \leq m$ **then**
**7**         **if** $u < \frac{a}{a+b}$ **then**
**8**             $a_i \leftarrow a + b$ ; $a_{\text{pendingIndex}} \leftarrow 0$ ; pendingIndex $\leftarrow i$
**9**         **else**
**10**            $a_i \leftarrow 0$ ; $a_{\text{pendingIndex}} \leftarrow a + b$
**11**         **end**
**12**     **else**
**13**         **if** $u < \frac{m-b}{2m-a-b}$ **then**
**14**            $a_i \leftarrow m$ ; $a_{\text{pendingIndex}} \leftarrow a + b - m$
**15**         **else**
**16**            $a_i \leftarrow a + b - m$ ; $a_{\text{pendingIndex}} \leftarrow m$ ; pendingIndex $\leftarrow i$
**17**         **end**
**18**     **end**
**19 end**
**20 return** $a_1, a_2, \ldots, a_n$

---

rest will be equal to 0, and for all $i \in \{1, \ldots, n\}$, $\mathbb{E}[\tilde{a}_i] = a_i$. When $m = 1$, we call this algorithm `round-pr` and when $a_1, \ldots, a_n$ are natural numbers we call it `round-nat`.

*Proof.* We begin by showing that for all $i \in [n]$, $\mathbb{E}[\tilde{a}_i] = a_i$. At step $i$, we have two numbers $a$ and $b$ which get rounded and obtain new values denoted by $\tilde{a}$ and $\tilde{b}$, respectively. The rounding depends on the value of $a + b$. If $a + b$ is less than or equal to $\beta$, then $\mathbb{E}[\tilde{a}] = (a + b)\frac{a}{a+b} = a$. If $a + b$ is greater than $\beta$ and at most $2\beta$, then $\mathbb{E}[\tilde{a}] = v\frac{\beta-b}{2\beta-a-b} + (a + b - \beta)\frac{\beta-a}{2\beta-a-b} = a$. Similarly, we obtain $\mathbb{E}[\tilde{b}] = b$. Since the expected values remain constant throughout the process, we conclude that for all $i \in [n]$, $\mathbb{E}[\tilde{a}_i] = a_i$.

We notice that for any step $i$ of the algorithm and for all $j$ that are smaller than $i$ but are not the pendingIndex, $a_j = 0$ or $a_j = \beta$. As a result, at the end of the algorithm all elements with index $j$ such that $j < n$ and $j \neq$ pendingIndex and one of the $n$-th or the pendingIndex elements are rounded to either 0 or $\beta$. The remaining element is the only one that can have a value in all $[0, \beta]$. Since $\sum_{i=1}^{n} a_i =$ sum, $\lfloor \frac{\text{sum}}{\beta} \rfloor$ elements are $\beta$ and if the remainder sum $- \lfloor \frac{\text{sum}}{\beta} \rfloor \beta$ is non-zero, it is assigned to the remaining element. $\square$

### A.4 Proof of Corollary 2.1

**Corollary 2.1.** If $m = 1$ and $\sum_{i=1}^{n} a_i = k \in \mathbb{N}$, then after the dependent rounding of `round-pr` $k$ elements will be equal to 1 and the rest will be equal to 0.

*Proof.* The proof follows directly from Lemma 2.2. $\square$

# B   Proofs from Section 3

## B.1   Proof of Theorem 3.1

**Theorem 3.1.** For any classifier $C$ that assigns scores $s_1, \ldots, s_n$ to $n$ candidates, Algorithm 1 solves the fairness-preserving cohort selection problem by selecting $k$ candidates with marginal probabilities $p_1, \ldots, p_n$ which achieve optimal linear utility $\sum_{i=1}^{n} p_i s_i$.

*Proof.* Algorithm 1 consists of two parts. In the first part, it modifies the input scores so that they sum up to $k$, which is the size of the cohort, and for the second part it randomly selects $k$ individuals by calling the `round-pr` subroutine. We can simplify the first part by rewriting it in a more concise way. If the sum of $C$'s scores $s_1, \ldots, s_n$ is less than $k$, then for any candidate $i \in [n]$ we set $p_i = \min\{s_i + b, 1\}$, with $b$ in $[0, 1]$ such that $\sum_{i=1}^{n} p_i = k$. More specifically, $p_i$ are initially set to $s_i + c$ so that their sum is equal to $k$. Their sum will remain equal to $k$, because the following loop only redistributes probability mass among candidates. If at any iteration some candidate $i$ has $p_i > 1$, then their marginal selection probability is set to 1 and the excess probability mass is redistributed evenly to all candidates with marginal probabilities less than 1. At the end, any probability $p_i$ that is less than 1 is equal to $s_i$ plus a constant $b$ that consists of the initial $c$ and the fractions of the probabilities that exceeded 1. Similarly, we have that if the sum of $C$'s scores is greater than $k$, then for any candidate $i$, $p_i = \max\{s_i - b, 0\}$ for a real number $b \in [0, 1]$ such that $\sum_{i=1}^{n} p_i = k$. If the sum is equal to $k$, the cohort selection marginal probabilities are equal to $C$'s scores. Finally, by Lemma 2.2 we obtain that for any $i \in [n]$ the expected value of indicator $\tilde{p}_i$ is $p_i$, and since $\tilde{p}_i$ is either 0 or 1, the $i$-th candidate is selected with probability $p_i$.

First, we want to show that the cohort formed by Algorithm 1 preserves fairness. Depending on the value of $\sum_{i=1}^{n} s_i$, we have three cases.

1. $\sum_{i=1}^{n} s_i = k$. The input scores are used unaltered as the marginal probabilities for the cohort selection and, thus, we have that for any pair of indices $i, j$, $|p_i - p_j| = |s_i - s_j|$.

2. $\sum_{i=1}^{n} s_i < k$. The cohort selection marginal probabilities are of the form $p_i = \min\{s_i + b, 1\}$. For any pair of individuals $i, j$, if both have probability of being selected 1, then $|p_i - p_j| = 0$. Else, if $p_i = 1$ and $p_j = s_j + b$, we have $|p_i - p_j| < |s_i - s_j|$ because $s_i + b \geq 1$. Finally, if $p_i = s_i + b$ and $p_j = s_j + b$, then it holds that $|p_i - p_j| = |s_i - s_j|$. As a result, we conclude that for any pair of individuals $i, j$, we have $|p_i - p_j| \leq |s_i - s_j|$.

3. $\sum_{i=1}^{n} s_i > k$. In this case, the cohort selection probabilities are of the form $p_i = \max\{s_i - b, 0\}$. For any pair of individuals $i, j$, if both have zero probability of being selected then $|p_i - p_j| = 0$. Else, if $p_i = 0$ and $p_j = s_j - b$, we have $|p_i - p_j| < |s_i - s_j|$. Finally, if $p_i = s_i - b$ and $p_j = s_j - b$, then it holds that $|p_i - p_j| = |s_i - s_j|$. Thus, for any pair of individuals $i, j$, we have $|p_i - p_j| \leq |s_i - s_j|$.

Second, we show that this cohort optimizes linear utility. The fairness-preserving solution that optimizes linear utility solves the linear program defined in Section 1.1 and satisfies the first constraint with equality, i.e. $\sum_{i=1}^{n} p_i = k$. This holds because otherwise we would be able to increase $p_i$ by setting $p_i' = \min\{p_i + d, 1\}$, for some $d \in (0, 1]$, so that $\sum_{i=1}^{n} p_i' = k$ and obtain greater utility while not violating any constraint.

We assume that there exists a different fairness-preserving solution $p_1', \ldots, p_n'$ which achieves higher utility, i.e. $\sum_{i=1}^{n} p_i' s_i > \sum_{i=1}^{n} p_i s_i$. Without loss of generality we can assume that the original probabilities $s_i$ are sorted in increasing order. Specifically, we have $s_1 \leq s_2 \leq \ldots \leq s_n$. The solution of Algorithm 1 maintains the same order, i.e. $p_1 \leq p_2 \leq \ldots \leq p_n$. Let $i$ be the individual with the smallest index for whom the two solutions differ. There are two possible cases. If $p_i' < p_i$, then since $\sum_{i=1}^{n} p_i = \sum_{i=1}^{n} p_i' = k$ there exists an individual with $j > i$ such that $p_j' > p_j$. In addition, we see that $p_i > 0$ and $p_j < 1$. The values of the probabilities depend on the sum of the input scores in comparison to $k$. If the sum is greater than $k$, all $p_i$ that are not zero are of the form $p_i = s_i - b$. Hence, considering that $p_i' < p_i \leq p_j < p_j'$ we obtain $|p_i' - p_j'| = p_j' - p_i' > s_j - b - (s_i - b) = s_j - s_i = |s_i - s_j|$. Similarly, if the sum is less than $k$, the values of $p_i$ that are not one are of the form $p_i = s_i + b$, therefore giving $|p_i' - p_j'| = p_j' - p_i' > s_j + b - (s_i + b) = s_j - s_i = |s_i - s_j|$.

If the sum is equal to $k$, we have $|p'_i - p'_j| = p'_j - p'_i > p_j - p_i = s_j - s_i = |s_i - s_j|$. In all three cases the constraints of the optimization are violated.

If $p'_i > p_i$, then there exists an individual with $j > i$ such that $p'_j < p_j$. Let $j$ be the smallest such index. If there exists another index $l > j$ such that $p'_l \geq p_l$, then by the previous argument the constraints are not satisfied. Therefore, for any candidate $l$ after $j$ $p'_l < p_l$. This solution achieves non-optimal utility because there exist candidates with $h < l$ and $p'_h > p_h$, such that we can move probability mass from candidates $h$ to individuals $l$ without violating the constraints. By constructing a solution $p''_1, \ldots, p''_n$ which gives a greater utility than $p'_1, \ldots, p'_n$, we arrive at a contradiction. Therefore, Algorithm 1 finds the optimal fairness-preserving solution. $\qquad\square$

## B.2 Proof of Corollary 3.1

**Corollary 3.1.** If $C$ is individually fair, then Algorithm 1 is individually fair.

*Proof.* The individual fairness condition for Algorithm 1 is satisfied by the assumption that the input scores are individually fair. More specifically, if the individual scores from $C$ are individually fair with respect to a metric $\mathcal{D}$, we obtain that $\forall i, j \in [n], |p_i - p_j| \leq |s_i - s_j| \leq \mathcal{D}(i, j)$. $\qquad\square$

## B.3 Proof of Lemma 3.1

**Lemma 3.1.** Given selection scores $s_1, \ldots, s_n$ from $C$, for a set of $n$ candidates and a parameter $\varepsilon \in (0, 1]$, we split $(0, 1]$ into $m = \lceil \frac{1}{\varepsilon} \rceil$ intervals of length $\varepsilon$ and for all $i \in [n]$ we set $\hat{s}_i = g\varepsilon$, where $g$ is such that $s_i \in ((g-1)\varepsilon, g\varepsilon]$. When Algorithm 1 runs for input $\hat{s}_1, \ldots, \hat{s}_n$ and cohort size $k$ it solves the $\varepsilon$-approximate fairness-preserving cohort selection problem with marginal selection probabilities $p_1, \ldots, p_n$, and it achieves linear utility $\sum_{i=1}^{n} p_i s_i \geq \sum_{i=1}^{n} p_i^* s_i - k(\varepsilon + 2\sqrt{\varepsilon})$, where $p_1^*, \ldots, p_n^*$ is the optimal solution for the offline fairness-preserving problem for input $s_1, \ldots, s_n$.

*Proof.* We have that for all individuals $i$ in $[n]$, $0 \leq \hat{s}_i - s_i \leq \varepsilon$. Therefore, we obtain that for any pair of individuals $i, j$, $|\hat{s}_i - \hat{s}_j| \leq |s_i - s_j| + \varepsilon$. By Theorem 3.1 we have that $|p_i - p_j| \leq |\hat{s}_i - \hat{s}_j|$. Combining the two inequalities, we obtain that fairness is preserved $\varepsilon$-approximately.

Note that Algorithm 1 always adjusts the scores of candidates such that its final output probabilities sum to $k$. For any candidate $i$ we observe that since the rounded score $\hat{s}_i$ is greater than $s_i$ by at most $\varepsilon$, the two output probabilities, $p_i$ and $p_i^*$ are also close, at most $\varepsilon$ apart. Depending on the sum of the input scores there are three cases.

1. If $\sum_{i=1}^{n} s_i \leq k$ and $\sum_{i=1}^{n} \hat{s}_i \leq k$, then Algorithm 1 shifts all the probabilities up. More specifically, we have that $p_i^* = \min\{1, s_i + x\}$ and $p_i = \min\{1, \hat{s}_i + y\}$, for some non-negative $x$ and $y$. Suppose that $y > x$, then for any candidate $i$ whose marginal selection probability $p_i$ has not been clipped to 1, $p_i = \hat{s}_i + y > s_i + x = p_i^*$. By summing up for all candidates we obtain that $k = \sum_{i=1}^{n} p_i^* < \sum_{i=1}^{n} p_i = k$, which is not feasible. Similarly, suppose that $x > y + \varepsilon$, then for any candidate $i$ whose marginal selection probability $p_i^*$ is has not been clipped, $p_i^* = s_i + x > \hat{s}_i + y = p_i$. As a result, we get that $\sum_{i=1}^{n} p_i^* > \sum_{i=1}^{n} p_i$, which is again false. Hence, we see that $y \leq x \leq y + \varepsilon$.

2. If $\sum_{i=1}^{n} s_i \geq k$ and $\sum_{i=1}^{n} \hat{s}_i \geq k$, then Algorithm 1 shifts all the probabilities down, so that both sums become equal to $k$. Thus, $p_i^* = \max\{0, s_i - x\}$ and $p_i = \max\{0, \hat{s}_i - y\}$, for some non-negative $x$ and $y$. Suppose that $y < x$, then for any candidate $i$ who has non-zero marginal selection probability $p_i^*$, we have that $p_i^* = s_i - x < \hat{s}_i - y = p_i$. By summing up for all candidates we obtain that $k = \sum_{i=1}^{n} p_i^* < \sum_{i=1}^{n} p_i = k$, which is not feasible. Similarly, suppose that $y > x + \varepsilon$, then for any candidate $i$ whose marginal selection probability $p_i$ is not 0, $p_i = \hat{s}_i - y < s_i - x = p_i^*$. As a result, we get that $\sum_{i=1}^{n} p_i^* > \sum_{i=1}^{n} p_i$, which is false. Thus, we see that $x \leq y \leq x + \varepsilon$.

3. If $\sum_{i=1}^{n} s_i \leq k$ and $\sum_{i=1}^{n} \hat{s}_i \geq k$, then Algorithm 1 sets $p_i^* = \min\{1, s_i + x\}$ and $p_i = \max\{0, \hat{s}_i - y\}$, for some non-negative $x$ and $y$. Suppose that $x + y > \varepsilon$, then for any candidate $i$ who has non-zero marginal selection probability $p_i$, we have that $p_i = \hat{s}_i - y < s_i + x$ and $p_i < 1$; therefore $p_i < p_i^*$.

By summing up the selection probabilities of all the candidates we obtain that $k = \sum_{i=1}^{n} p_i < \sum_{i=1}^{n} p_i^* = k$, which is not feasible. Similarly, suppose that $x + y < 0$, then for any candidate $i$ whose marginal selection probability $p_i^*$ is not 1, $p_i^* = s_i + x < \hat{s}_i - y = p_i^*$. As a result, we get that $\sum_{i=1}^{n} p_i^* < \sum_{i=1}^{n} p_i$, which is false. Therefore, we know that $0 \leq x + y \leq \varepsilon$.

In all three cases we see that for any candidate $i$ the selection probabilities of the two solutions satisfy $|p_i - p_i^*| \leq \varepsilon$. Now, consider any coordinate $i$ for which $p_i^* \geq \sqrt{\varepsilon}$. Since $p_i \geq p_i^* - \varepsilon$, we have that $p_i \geq (1 - \sqrt{\varepsilon})p_i^*$ and, thus, $p_i s_i \geq (1 - \sqrt{\varepsilon})p_i^* s_i$. Let $E$ be the set of all individuals who have $p_i^* < \sqrt{\varepsilon}$.

1. If $\sum_{i=1}^{n} s_i \leq k$, then $p_i^* = \min\{1, s_i + x\}$. Thus, $0 \leq s_i + x < \sqrt{\varepsilon}$ because both $s_i$ and $x$ are non-negative. Since both $p_i$ and $p_i^*$ sum up to $k$, we obtain that $-kx \leq \sum_{i \in E} p_i s_i \leq k\sqrt{\varepsilon} - kx$ and $-kx \leq \sum_{i \in E} p_i^* s_i \leq k\sqrt{\varepsilon} - kx$. Therefore, $\sum_{i \in E} p_i s_i \geq \sum_{i \in E} p_i^* s_i - k\sqrt{\varepsilon}$.

2. If $\sum_{i=1}^{n} s_i \geq k$ and $\sum_{i=1}^{n} \hat{s}_i \geq k$, then $p_i^* = \max\{0, s_i - x\}$ and $p_i = \max\{0, \hat{s}_i - y\}$. Thus, $s_i - x < \sqrt{\varepsilon}$. Furthermore, both solutions assign zero probability to candidates with $s_i - x < -\varepsilon$. As a result, all individuals in $E$ whose $s_i - x > -\varepsilon$, have $s_i$ within an interval of length $\sqrt{\varepsilon} + \varepsilon$. Similarly to case 1, we have that $\sum_{i \in E} p_i s_i \geq \sum_{i \in E} p_i^* s_i - k(\sqrt{\varepsilon} + \varepsilon)$.

If we split the linear utility to the utility of individuals in $E$ and not in $E$, we see that

$$\sum_{i=1}^{n} p_i s_i = \sum_{i \notin E} p_i s_i + \sum_{i \in E} p_i s_i \geq (1 - \sqrt{\varepsilon}) \sum_{i \notin E} p_i^* s_i + \sum_{i \in E} p_i^* s_i - k(\varepsilon + \sqrt{\varepsilon}) \geq \sum_{i=1}^{n} p_i^* s_i - k(\varepsilon + 2\sqrt{\varepsilon})$$

$\square$

## B.4 Proof of Theorem 3.2

**Theorem 3.2.** For any classifier $C$ that assigns scores $s_1, \ldots, s_n$ to $n$ candidates, Algorithm 2 solves the online $\varepsilon$-approximate fairness-preserving cohort selection problem for any $\varepsilon \in (0, 1]$ by selecting a cohort of size $k$ with marginal probabilities $p_1, \ldots, p_n$, achieves linear utility $\sum_{i=1}^{n} p_i s_i \geq \sum_{i=1}^{n} p_i^* s_i - k(\varepsilon + 2\sqrt{\varepsilon})$ (where $p_1^*, \ldots, p_n^*$ is the optimal solution for the offline fairness-preserving problem for input $s_1, \ldots, s_n$), and keeps at most $O(k + \frac{1}{\varepsilon})$ candidates pending.

*Proof.* We want to prove that Algorithm 2 and the algorithm described in Lemma 3.1 compute the same solution. In other words, we show that the probability $p_i$ that individual $i$ is selected by Algorithm 2 is equal to the probability $q_i$ that $i$ is selected by Algorithm 1 applied to the modified individual scores $\hat{s}_1, \ldots, \hat{s}_n$, where $\hat{s}_i = g\varepsilon$ if $s_i \in ((g-1)\varepsilon, g\varepsilon]$. The first step is to prove that the scores $\tilde{s}_1, \tilde{s}_2, \ldots, \tilde{s}_n$ which are the input of the final rounding in line 28 satisfy the following properties:

1. $\sum_{i=1}^{n} \tilde{s}_i = k$

2. $\mathbb{E}[\tilde{s}_i] = q_i, \forall i \in [n]$.

We saw in the proof of Theorem 3.1 that for the offline algorithm all candidates of the same group have the same selection probability $q_i$. We will show that for any person $i$ we have $q_i = s^g$, where $s^g$ is the final calculated probability of any member of group $g$ to which $i$ belongs. The $s^g$ we are referring to is calculated by the final execution of lines 6-22. These lines of Algorithm 2 describe the same procedure as lines 2-15 of Algorithm 1, but from the point of view of groups instead of individual candidates. We consider three cases that depend on the value of the sum of scores from $C$.

**Case 1**: $\sum_{i=1}^{n} \hat{s}_i = k$. The offline algorithm considers the scores as selection probabilities and, thus, assigns probability $q_i = \hat{s}_i = g\varepsilon$ to candidate $i$ of group $g$. Similarly, Algorithm 2 assigns probability $s^g = g\varepsilon$ to group $g$ and in line 27 sets $\tilde{s}_i = n_i s^g = n_i g\varepsilon = n_i q_i$ for the people who have not been rejected.

**Case 2**: $\sum_{i=1}^{n} \hat{s}_i < k$. The process that calculates $s^g$ starts by setting $s^g = g\varepsilon + c$. The offline algorithm initializes the probability of $i$ who is a member of $g$ as $q_i = \hat{s}_i + c = g\varepsilon + c$. If for all groups $g$, $g\varepsilon + c \leq 1$, then the adjustment stops for both algorithms and we have that for any group $g$, any member $i$ of $g$ has $q_i = s^g$. If there exists $g$ such that $q_i > 1$, then the corresponding $s^g$ exceeds 1 by the same amount. Additionally, at this point the probabilities of all people in the same group as $i$ will exceed 1 by this amount. Therefore, the $n_{<1}$ of the two algorithms is the same. The offline algorithm runs the loop for all members of all groups that have individuals with $q_i > 1$. The online version aggregates the excess mass from all $n^g$ members of group $g$ and redistributes it all at once instead of running separate iterations for every member of the group as the offline does. No extra mass is added after the initialization but it is only moved from group to group. Hence, we obtain that at the end of this process $q_i = s^g$.

**Case 3**: $\sum_{i=1}^{n} \hat{s}_i > k$. Similar to case 2, if candidate $i$ is a member of group $g$, then $q_i = s^g$. Those who were rejected have $\tilde{s}_i = 0$ and $n_i = 0$. As a result, we see that for any person $i$, $\tilde{s}_i = n_i q_i$. From this we can infer that

$$\sum_{i=1}^{n} \tilde{s}_i = \sum_{i=1}^{n} n_i q_i = \sum_{g=0}^{m} \sum_{i \in g} n_i s^g = \sum_{g=0}^{m} n^g s^g = \sum_{g=0}^{m} \sum_{i \in g} q_i = k.$$

As new people are added to the groups, the group probabilities $s^g$ become smaller in order for the sum of probabilities $n^g s^g$ to be equal to $k$. Therefore, the maximum number $v$ of people that can be represented by a candidate in a given group either stays the same or increases after every iteration. By Lemma 2.2, we obtain that the rounding process maintains the expected value of the number of people each candidate represents equal to their initial value. Since every person begins by representing only themselves, we have that for the $i$-th candidate $\mathbb{E}[n_i] = 1$. Finally, we obtain $\mathbb{E}[\hat{s}_i] = \mathbb{E}[n_i q_i] = \mathbb{E}[n_i]q_i = q_i$, because the calculation of $s^g$s is deterministic. The final rounding procedure makes the final decisions and outputs 0 if the candidate is rejected and 1 if the candidate is selected. Due to properties 1 and 2, the probability of candidate $i$ being selected by the online algorithm is $q_i$. Thus, Algorithm 2 and the offline algorithm with input scores rounded to multiples of $\varepsilon$ have the same selection probabilities. By Lemma 3.1, Algorithm 2 preserves fairness $\varepsilon$-approximately and achieves linear utility $\sum_{i=1}^{n} p_i s_i \geq \sum_{i=1}^{n} p_i^* s_i - k(\varepsilon + 2\sqrt{\varepsilon})$, where $p_1^*, \ldots, p_n^*$ is the optimal solution for the offline fairness-preserving problem for input $s_1, \ldots, s_n$.

Because of the online probability adjustments, we have that for $n \geq k$ the sum of all the probabilities is equal to $k$ at the end of each loop. Therefore, we have $\sum_{g=0}^{m} n_g s^g = k$. If $s^f > 0$, each person can represent at most $\lfloor \frac{1}{s^g} \rfloor$ candidates. By Lemma 2, the number of representatives per group is at most

$$\left\lceil \frac{n^g}{\lfloor \frac{1}{s^g} \rfloor} \right\rceil \leq \frac{n^g}{\lfloor \frac{1}{s^g} \rfloor} + 1 \leq 2n^g s^g + 1,$$

since $\frac{n_g}{\lfloor \frac{1}{s^g} \rfloor} \leq 2n_g s^g$. If we sum up the number of representatives for all groups we obtain

$$\sum_{g=0}^{m} \left\lceil \frac{n^g}{\lfloor \frac{1}{s^g} \rfloor} \right\rceil \leq \sum_{g=0}^{m} (2n^g s^g + 1) = 2k + \left\lceil \frac{1}{\varepsilon} \right\rceil + 1.$$

This completes the proof. □

## B.5 Proof of Corollary 3.2

**Corollary 3.2.** If $C$ is individually fair, then Algorithm 2 is $\varepsilon$-individually fair.

*Proof.* If the individual probabilities of selection by $C$ are individually fair with respect to a metric $\mathcal{D}$, we obtain that $\forall i, j \in [n]$

$$|p_i - p_j| \leq |s_i - s_j| + \varepsilon \leq \mathcal{D}(i, j) + \varepsilon.$$

□

## C  Proofs from Section 4

### C.1  Proof of Lemma 4.1

**Lemma 4.1.** If $sum < k$, Algorithm 3 increases each input score $s_i$ to a final output $p_i = s_i + \alpha_i \geq s_i$ such that all of the candidates $j$ with $p_j < 1$ receive the same cumulative adjustment value $\alpha_j = v$.

*Proof.* First we observe that $p_i \geq s_i, \forall i$. At each step of the algorithm the value of $p_i$ either increases, or is clipped at 1. Since $s_i \in [0,1], \forall i$, we therefore know that $p_i \geq s_i$.

Algorithm 3 initially adds a constant amount $c$ to every candidate, then repeatedly redistributes overflow evenly across all candidates with $s_i + \alpha_i < 1$. We prove by induction that for all candidates $i$ with $p_i < 1$, their increment $\alpha_i$ are the same. Consider two candidates $i, j$ such that their final $p_i, p_j < 1$. As argued above, both $p_i, p_j$ are smaller than 1 throughout the execution of the algorithm. At the beginning, $\alpha_i = \alpha_j = c$ so the claim holds.

Given that these candidates are equal at a certain step $d$, we also know they will remain equal at the next step $d+1$. By assumption, neither candidate reaches 1 on either of these steps; therefore they will both receive an adjustment. When an adjustment is made, all candidates are affected equally. $\qquad\square$

### C.2  Proof of Theorem 4.1

**Theorem 4.1.** For any classifier $C$ that assigns scores $s_1, \ldots, s_n$ to $n$ candidates, Algorithm 3 solves the fairness-preserving cohort selection problem by selecting $k$ candidates with marginal probabilities $p_1, \ldots, p_n$ that achieve the optimal value of ratio utility $\min_i \frac{p_i}{s_i}$.

*Proof.* We first show that Algorithm 3 either preserves or decreases the difference between any pair of individuals. Let $p_i$ denote the probability that candidate $i$ is selected by Algorithm 3. Thus we must show, for all $i, j$, $|p_i - p_j| \leq |s_i - s_j|$. We apply Algorithm 3 to a set of candidates and obtain two disjoint sets of output candidates: let $T_{(=1)}$ be the set of candidates whose output $p_i = 1$, and let $T_{(<1)}$ be the set of candidates whose output $p_i < 1$. Let $T_{all}$ be the union of these two disjoint sets. The algorithm behaves differently depending on the value of $\sum_i s_i$.

1. First, consider when $\sum_i s_i < k$. Let $\alpha_i$ be the cumulative adjustment received by candidate $i$ during the algorithm. The adjustments are exactly chosen such that $\sum_i (s_i + \alpha_i) = k$. Then we apply `round-pr`, where we know by Lemma 2.2 that the probability of being selected is preserved during rounding $\Pr[\text{round-pr} \text{ selects } i] = p_i = s_i + \alpha_i$ We thus need to cover three cases:

   (a) First, we consider two elements $s_i, s_j \in T_{(=1)}$, $|p_i - p_j| = |1 - 1| \leq |s_i - s_j|$.
   (b) Next, we have $s_i, s_j \in T_{(<1)}$ By Lemma 4.1, these candidates all receive the same adjustment. Let this adjustment be called $b$, $|p_i - p_j| = |s_i + b - (s_j + b)| = |s_i - s_j|$.
   (c) Finally we have $s_i \in T_{(=1)}, s_j \in T_{(<1)}$. Note that $\alpha_i \leq \alpha_j$, and $s_i > s_j$. We see that $|p_i - p_j| = |s_i + \alpha_i - (s_j + \alpha_j)| = |s_i - s_j + \alpha_i - \alpha_j| \leq |s_i - s_j|$.

2. Next, consider $\sum_i s_i \geq k$. Here we know $\frac{k}{\sum_i s_i} \leq 1$. By Lemma 2.2, $p_i = \frac{k}{\sum_l s_l} s_i$. Then, for candidates $i$ and $j$, we have:

$$|p_i - p_j| = \left| \frac{k}{\sum_l s_l} s_i - \frac{k}{\sum_l s_l} s_j \right| = \left| \frac{k}{\sum_l s_l} (s_i - s_j) \right| \leq |s_i - s_j|$$

Thus we see that $|s_i - s_j| \geq |p_i - p_j|$.

We next show that among fairness-preserving solutions to the cohort selection problem, Algorithm 3 achieves the maximum value of $\min_i \frac{p_i}{s_i}$. Let $T_{(<1)}$ and $T_{(=1)}$ be defined as above, and note that $T_{(=1)}$ may be empty. Consider for contradiction an arbitrary algorithm $A'$ with probabilities of selection $p'_i$ which also preserves fairness while achieving better utility: $\min_i \frac{p'_i}{s_i} > \min_i \frac{p_i}{s_i}$.

1. First, consider the case where $\sum_i s_i < k$ and $T_{(=1)}$ is nonempty. Let individual $j$ have the most extreme ratio: $\frac{p_j}{s_j} = \min_i \frac{p_i}{s_i}$. Consider how to make this ratio minimal. For all individuals in $T_{(<1)}$, increasing $s_i$ will reduce this ratio: $\frac{p_i}{s_i} = \frac{s_i+b}{s_i} = 1 + \frac{b}{s_i}$. If any individuals $j$ are in $T_{(=1)}$, they will have received some adjustment $\alpha_j \leq b$, and thus they will have an even smaller ratio. Therefore, we know that individual $j$ must have $s_j \geq s_i, \forall i$ and be a member of $T_{(=1)}$, and we have that for Algorithm $A$, $\min_i \frac{p_i}{s_i} = \frac{p_j}{s_j} = \frac{1}{s_j}$.

   By our definition of $A'$, there is some potentially different most extreme element $l$ providing better utility: $\frac{p'_l}{s_l} = \min_i \frac{p'_i}{s_i} > \frac{p_j}{s_j}$.

   (a) If $s_l = s_j$, the largest minimum ratio $\frac{p'_l}{s_l}$ that $A'$ can achieve is $\frac{1}{s_l} = \frac{1}{s_j}$, which is the same ratio as Algorithm A and contradicts our assumption.

   (b) If $s_l \neq s_j$, we know $s_l < s_j$, since we already know that $s_j$ is at least as big as all other elements. Since $p_j$ is 1, we know that $\frac{p_j}{s_j} \geq \frac{p'_j}{s_j} > \frac{p'_l}{s_l}$. This contradicts our assumption that $A'$ has greater utility than $A$.

2. Next, consider the case where $\sum_i s_i < k$ and $T_{(=1)}$ group is empty. Again, let element $j$ be the most extreme element for algorithm $A$, i.e. $\min_i \frac{p_i}{s_i} = \frac{p_j}{s_j}$. Consider $A'$ on its most extreme element $l$ constructed as in Item 1.

   (a) This time, if $s_l = s_j$, we may have $p'_l > p_l$, achieving greater utility. To preserve the distances, if $A'$ adjusts element $l$ by a certain amount, it must adjust all smaller elements by at least the same amount. However, we know that $\sum_i p_i = k$, so this required extra adjustment will mean that $\sum_i p'_i > k$, and $A'$ will not be selecting exactly $k$ candidates.

   (b) If $s_l < s_j$, then person $j$ has the largest score, and similar to (2a), if $p'_j > p_j$ we see that $\sum_i p'_i > k$. If $p'_j \leq p_j$ we have the contradiction in (1b).

3. Finally, consider the case where $\sum_i s_i \geq k$. In Algorithm 3, all elements will be multiplied by a factor of $\frac{k}{\sum_i s_i}$, and then by Lemma 2.2, any element will satisfy $p_i = \frac{k}{\sum_l s_l} s_i$, so we have $\min_i p_i/s_i = \min_i \left( \frac{k}{\sum_l s_l} s_i \right)/s_i = \frac{k}{\sum_i s_i}$ Notice that in this case, all elements are adjusted by the same constant ratio. Using extreme element $l$ for algorithm $A'$ as constructed previously, and again supposing that this element is more extreme than any element in algorithm $A$, we derive that $p'_l > s_l \frac{k}{\sum_i s_i}$. Since element $l$ was the minimum ratio for algorithm $A'$, it follows that for all $i$, $p'_i > s_i \left( \frac{k}{\sum_i s_i} \right)$. Taking the sum on both sides we see $\sum_i p'_i > \sum_i s_i \left( \frac{k}{\sum_i s_i} \right) = k$. As before, the number of elements chosen by algorithm $A'$ will be more than $k$, which is a contradiction.

Thus we conclude that $A'$ cannot exist. $\qquad\square$

### C.3 Proof of Corollary 4.1

**Corollary 4.1.** If $C$ is individually fair, then Algorithm 3 is individually fair.

*Proof.* The proof is analogous to that of Corollary 3.1. $\qquad\square$

### C.4 Proof of Theorem 4.2

**Theorem 4.2.** For any classifier $\mathcal{C}$ that assigns scores $s_1, \ldots, s_n$ to $n$ candidates, Algorithm 4 solves the fairness-preserving cohort selection problem by selecting $k$ candidates with marginal probabilities $p_1, \ldots, p_n$ that achieve the optimal value of ratio utility. The algorithm leaves no more than $O(k)$ candidates pending at any time.

*Proof.* We first show that, for a list of candidate scores $s_1, \ldots, s_n$, the selection probability given by the offline Algorithm 3 for any candidate $i$ is the same as that given by Algorithm 4, and therefore provides an optimal utility solution to the problem. Let Algorithm 4 be denoted $\mathcal{A}$ and let $\mathcal{A}$ have probability $p_i$ of selecting candidate $i$. Let Algorithm 3 be denoted $\mathcal{B}$ with probability $q_i$ selecting candidate $i$. $\mathcal{A}$ never accepts candidates until the end of the stream, thus there are two major cases.

1. The stream ends with sum $\geq k$, and $\mathcal{A}$ selects all candidates in PENDING. A candidate $i$ is added to PENDING in one of three ways:

   (a) Candidate $i$ was in TOP when the sum reached $k$. At each round after the sum exceeded $k$, they must survive `round-pr`, which by Lemma 2.2 always preserves their marginal probability. Therefore we only need to consider the effect of the incremental scaling adjustments. First, let $\text{scale}_t$, $\text{sum}_t$, and $\text{incr}_t$, represent the values of these variables at some step $t$ when the sum first exceeds $k$. We initialized $\text{scale}_t \leftarrow \frac{k}{\text{sum}_t}$. Consider the value of $\text{scale}_{t+1}$:

   $$\text{scale}_{t+1} = \text{incr}_t \cdot \text{scale}_t = \frac{\text{sum}_{t+1} - s_{t+1}}{\text{sum}_{t+1}} \cdot \text{scale}_t = \frac{\text{sum}_t}{\text{sum}_{t+1}} \cdot \frac{k}{\text{sum}_t} = \frac{k}{\text{sum}_{t+1}}$$

   Thus, we can see by induction that for all time steps $t$, $\text{scale}_t = \frac{k}{\text{sum}_t}$. Furthermore, we can express the final value of a single element $s_i$, which by definition is at position $i$ in the stream, as:

   $$p_i = s_i \cdot \text{scale}_i \cdot \prod_{t=i+1}^{n} \text{incr}_t = s_i \cdot \frac{k}{\text{sum}_i} \cdot \prod_{t=i+1}^{n} \frac{\text{sum}_t - s_t}{\text{sum}_t}$$
   $$= s_i \cdot \frac{k}{\text{sum}_i} \cdot \prod_{t=i+1}^{n} \frac{\text{sum}_{t-1}}{\text{sum}_t} = s_i \cdot \frac{k}{\text{sum}_n}$$

   Thus we see that the final $p_i$ is adjusted exactly the same way as it would be in the offline case.

   (b) Candidate $i$ was already in REST or arrived at the moment when the sum reached $k$. For each iteration, they must survive `round-pr`, which respects their marginal by Lemma 2.2. From that point forward, they are part of PENDING and treated the same as in 1a.

   (c) Candidate $i$ was encountered when sum $\geq k$. In this case, they are treated the same as in part 1a above.

   In either case, when $\mathcal{A}$ ends, a subset of candidates are selected from pending using `round-pr`, which again preserves their marginal probability. Thus $p_i = q_i$ in these cases.

2. The stream ends with sum $< k$. $\mathcal{A}$ has three sets at this point: TOP (the greatest $\lceil k/\alpha \rceil$ elements of the stream), REST (at most $\lceil k/\alpha \rceil$ elements rounded to $1 - \alpha$) and RAND (a randomly selected group of $k$ candidates, disjoint from TOP and REST).

   No acceptances are made until the stream ends. The top candidates are placed into TOP, and elements are added to REST via `round-pr`, which preserves original marginal probabilities exactly by Lemma 2.2. The only change in probabilities then comes from the additive adjustments made when the stream ends. The main difference between the additive adjustments here and the water-filling behavior in $\mathcal{B}$ is how we treat the people in RAND.

   For the purpose of proof, consider a conceptual algorithm $\mathcal{A}'$ that behaves exactly the same as $\mathcal{A}$, except instead of RAND it has ZEROS, a list of all candidates outside of TOP and REST. When this conceptual algorithm ends, all the mass from water-filling in ZEROS will be collected into a set of $k$ random candidates, which becomes the equivalent of the list RAND in $\mathcal{A}$.

   In more detail, at the end of $\mathcal{A}'$, each member of TOP, REST, and ZEROS is given $\frac{k - \text{sum}}{n}$, and any overflow is redistributed using water-filling. Any candidate $i$ now has the same probability of selection by both $\mathcal{A}'$ and $\mathcal{B}$: $s_i + \alpha_i$. $\mathcal{A}'$ then performs a final `round-pr` step to select the outputs. Thus, in $\mathcal{A}'$, TOP, REST, and ZEROS are given the same treatment they would receive in $\mathcal{B}$. Before the final step of selecting candidates by `round-pr`, $\mathcal{A}'$ collects all of the mass accumulated in ZEROS

from water-filling into a random subset of $k$ members from ZEROS. Thus the only difference between $\mathcal{A}$ and $\mathcal{A}'$ is that, $\mathcal{A}'$ first kept everyone and later uniformly sampled a subset of $k$ people, whereas $\mathcal{A}$ immediately keeps a random subset of $k$. This subsampling step does not change the marginal probabilities of any element in ZEROS; their probability before and after collecting the mass of ZEROS At the end, $\mathcal{A}$ selects candidates using `round-pr` on the entire set TOP $\cup$ REST $\cup$ RAND, which does not affect the adjusted marginals $s_i + \alpha_i$. Thus $p_i = s_i + \alpha_i = q_i \forall i$.

Next, we show that the algorithm keeps at most $O(k)$ candidates pending. Any candidate not in one of: TOP, REST, RAND, PENDING is considered rejected. The sizes of TOP and RAND are explicitly bounded by $\lceil k/\alpha \rceil$. REST is not bounded in size explicitly, but note that the set is maintained by constantly applying `round-pr`$(\{p_i\}_{i \in \text{REST}}, 1 - \alpha)$. After rounding, at most 1 person in REST has a value $< (1 - \alpha)$. If we ever have more than $\lceil k/(1 - \alpha) \rceil$ elements, we have $sum \geq \lceil k/(1 - \alpha) \rceil (1 - \alpha) > k$. This goes to the $sum \geq k$ case, which no longer adds elements to REST, thus it is bounded in length by $\lceil k/(1 - \alpha) \rceil$. PENDING is created initially by adding TOP and REST ($\lceil k/(1 - \alpha) \rceil + \lceil k/\alpha \rceil$ pending), but for subsequent steps is never longer than $k$ by Lemma 2.2. Thus the worst we can do is remain under a sum of k for the entire stream. TOP, REST, and RAND will all be kept pending until the end of the stream, but still we have at most $\lceil k/\alpha \rceil + \lceil k/(1 - \alpha) \rceil + \lceil k/\alpha \rceil = O(k)$ candidates pending. $\qquad \square$

### C.5  Proof of Corollary 4.2

**Corollary 4.2.** If $C$ is individually fair, then Algorithm 4 is individually fair.

*Proof.* The proof is analogous to that of Corollary 4.1. $\qquad \square$

