# OpenReview forum: "Fair and Useful Cohort Selection"
_TMLR — Accepted by TMLR_

### Review · Reviewer_zKWw · 2023-04-29

**Summary Of Contributions:**

The work considers the fair cohort selection problem, that is, selecting a set of $k$ individuals while satisfying individual fairness. The main contributions are
* A relaxation of the online cohort selection problem (when candidates arrive in an online fashion) that allows putting candidates 'on hold', rather than requiring an immediate response.
* High utility methods for selecting cohorts (online or offline) for two notions of utility. Linear (which matches the case where there exists a fair classifier with probability of selection closely correlated to the utility of each individual) and Ratio Utility (which aims to capture arbitrary utilities assigned to individuals).



**Audience:**

Yes

**Broader Impact Concerns:**

None - the work considers the problem of fair classification/cohort selection in the abstract.

**Claims And Evidence:**

Yes

**Requested Changes:**

**Requested Changes prior to recommendation**
* Clarify whether instances of "for any" in algorithms should be "for all".
* Clarify in the introduction that the relaxation for the offline setting does not necessarily immediately reject candidates.
* Clarify in the introduction whether the $u_i$ are assigned independently to each $i$ or if they can be a function of the whole cohort. (Presumably it is independent.)

**Recommended Changes to Strengthen the Work**
* Consider devoting more space/words to the new setting proposed for online cohort selection.
* Consider revising the algorithm notation and including inline comments to make them more readable and to match the prose descriptions provided.
* Consider restating lemmas/theorems in the appendix alongside the proofs, this will make them easier to follow.

**Strengths And Weaknesses:**

**Strengths**
1. The paper considers an important setting for fair classification and improves significantly on prior work.
2. The relaxation of online cohort selection to place candidates 'on hold' is an elegant path around the impossibility results. One suggestion for future work could be to consider whether there are methods that reduce the time on hold or ensure that similar candidates spend similar time on hold if additional opportunities to revisit candidates are intermixed, rather than held to the end of the stream.

**Weaknesses**
1. Although the relaxation for online cohort selection is nice, the reader is left wondering whether its complementary version (outputting acceptances throughout the stream and only rejecting at the end) would also work and whether this would be more realistic for some settings. For example, in hiring, the firm may want to make offers to candidates quickly to avoid the candidate accepting an offer from a different firm. Furthermore, it would be helpful to characterize whether similar candidates are left 'on hold' for similar amounts of time in the constructions presented (it seems that this might not hold for adversarial ordering).
2. The algorithms are difficult to parse and would be greatly improved with inline comments/pseudocode. For example, several constants are left without type, which can be inferred but interrupts the flow of reading. There are also places where it is unclear whether and how values are updated, for example, the $n_i$ seem to be updated as a result of the roundnat procedure, but this isn't clear from the notation. There are also several mismatches with the terminology in the text, for example, does "delete all $i$ ..." in line 26 of Algorithm 2 mean "reject"?

---

### Review · Reviewer_QhfX · 2023-05-03

**Summary Of Contributions:**

The paper starts by motivating the problem of composition in algorithmic fairness. It is known that composition of fair decisions may not lead to a fair composed algorithm, and this has been addressed in the context of dependent composition and cohort selection. The authors provide offline and online algorithms for cohort selection under two different utility setups.



**Audience:**

Yes

**Broader Impact Concerns:**

No concern here.

**Claims And Evidence:**

Yes

**Requested Changes:**

See weaknesses

**Strengths And Weaknesses:**

Strengths:
- This type of issue of fairness with composition is vastly understudied and I think it is nice that the authors are tackling some of the problems arising there. These types of settings where the decision we make on each individual also depends on the decision on other individuals are seldom studied (the usual framework looks at classifying/scoring people independently with no capacity constraint)
-There is something interesting about how the authors’ algorithms deal with the impossibility to obtain fairness in a fully online setting, by delaying decisions on a fraction of people and trading-off fairness vs how many people the algorithm has to put on “hold”.
- It is nice that the paper looks at several types of utilities, and in particular about not having full information about the utility of a candidate/agent.
- The algorithms work with a relatively small number of candidates pending for both ratio and linear utilities (of the order of k, the number of people selected in the cohort. In a hiring process, I would imagine that k is very small compared to the number of candidates, so this represent a small fraction of the population)

Weaknesses:
- Can Algorithm 1 be implemented through a linear program? If so, can the authors discuss a bit more how Algorithm 1 helps over solving the LP directly? Is it because it provides a basis for the online algorithm? It might be nice to highlight this.
- My main concerns however are with respect to the ratio utility:
i) The ratio utility is a bit weird. I understand why we care about $\sum_i p_i u_i$ where $u_i$ is unknown, I am not sure what the renormalization factor in the ratio represents, however. It would be nice for this to have more discussion. I understand $\sum_i s_i u_i$ as the utility we get if we use the scores directly as probability of selection, but why is that the right benchmark?
ii) Related to the above, the results for ratio utilities, starting from Lemma 2.1., seem to heavily rely on optimizing this specific ratio expression. This would not appear if for example we wanted to optimize just $\sum_i p_i u_i$. It’s also very worst case in a weird way; basically the worst case can only happen if the utility is 1 for a single agent and 0 for all others, which seems unrealistic. In fact, if the scores themselves were obtained through a reasonable process (not just fair, but also with some level of accuracy), there should be some correlation between $s_i$ and $u_i$ that would prevent such cases. It would be nice to develop this a bit more.

Overall, I think the main idea is interesting and the algorithms are nice, but I think the paper may need a bit more framing/justifying its assumptions before it is published. I think adding more justification and discussion of the ratio utility would be sufficient, in my opinion, to have this at TMLR!

---

### Review · Reviewer_3ZmY · 2023-06-05

**Summary Of Contributions:**

The paper studies the problem of cohort selection, i.e. selecting $k$ candidates out of $n$, which achieves both fairness and utility. This problem is studied in the offline and online full-information setting, and with linear and ratio utilities. Specifically, the approach considered is to compose predictions of a single classifier to select the cohort. The paper proposes two polynomial-time algorithms for offline settings and two utility functions, and their corresponding online versions. The interesting part of the algorithms are that they are individually fair if the classifier is individually fair. For the online algorithms, the interesting idea is to keep a queue of $O(k)$ candidates pending at any time for future comparison and fair selection.

**Audience:**

Yes

**Broader Impact Concerns:**

The proposed algorithms, specially the ones in the online setting, can be interesting to ensure fairness in online selection platforms. But without a numerical demonstration or a practical case study, it is hard to conclude about their applicability.

**Claims And Evidence:**

No

**Requested Changes:**

1. Can you add a justification against choosing the ratio utility? Where it will be useful in practice (by showing some theoretical or numerical results), or is it just a theoretical tool to prove the worst-case performance for any utility?
2. Can you show experimental results on some (real-world or real-like) selection dataset, where the proposed online or offline algorithms demonstrate benefit over existing algorithms for fair cohort selection? Specifically, given practicality of the problem under investigation and known experimental results in the literature, can you compare against existing individually meritocratic cohort selection techniques like [1] or collective meritocratic cohort selection techniques like [2]?
3. Can you please add results on the fairness of the proposed algorithms when the underlying classifier is only $\epsilon$-individually fair?

[1] Michael Kearns, Aaron Roth, and Zhiwei Steven Wu. Meritocratic fairness for cross-population selection. In International Conference on Machine Learning - Volume 70, pp. 1828–1836. 2017.

[2] Thomas Kleine Buening, Meirav Segal, Debabrota Basu, Anne-Marie George, and Christos Dimitrakakis. "On Meritocracy in Optimal Set Selection." In Equity and Access in Algorithms, Mechanisms, and Optimization, pp. 1-14. 2022.

**Strengths And Weaknesses:**

# Strengths

1. The paper addresses a timely problem of individually fair and utility maximising cohort selection while scores over the candidates are yielded from an individually fair classifier.

2. The authors uses ratio utility (a proxy of worst-case utility function) to show a case where the proposed method of Dwork & Ilvento (2019) fails to be utility maximising and the proposed approach triumphs.

3. The idea of using dynamic programming followed by dependent rounding constructs a clear and simple basis for the proposed algorithms.

4. The interesting algorithmic idea of this paper is to maintain a queue of pending applicants so that they can be considered in future while not growing the queue above $O(k)$.


# Weaknesses

1. The paper invents ratio utility as a proxy of the worst-case utility function and use it further to show benefits of the proposed method. I am not sure where it really matters in practice. Specifically,

i. it is not clear how does it correspond to the fair-utility maximising cohort selection problem under general individualistic utility, i.e. $\max \sum_i p_i u_i(s_1, ..s_n)$ with the constraints similar to the LP program in page 3.

ii. it is also not clear how does it correspond to the fair-utility maximising cohort selection under any collective utility, , i.e. $\max \sum_i p_i u_i(\mathrm{cohort}_k, s_1, ..s_n)$ as in Equation (2) in [1]. In brief, when the utility of an individual in the cohort depends on the other selected candidates.

A clear understanding of these relations, theoretically and/or numerically, would provide us a motivation to study the ratio utility. Right now, it is not clear.

2. Though the paper studies such a practical problem with existing literature in meritocratic fair selection/ranking, it is surprising that there is no numerical comparison or illustration of the goodness of this method. Without that, it is hard to accept that the proposed methods are effective in practice for general utility functions, as claimed.

3. The paper should at least numerically compare with the meritocratic fair selection and fair ranking algorithms. The justification that meritocratic fairness papers need scores to be reflective of 'true merit' is not agreeable. We can replace the $x_{ij}$ in [1] with scores and still the algorithm in [1] works. [2] shows that meritocratic fair selection can be generalised to include any utility function and any score representing a proxy of 'merit' to conduct fair and utility-maximising cohort selection. Also the argument against not comparing to [3] is weak if a numerical experimentation is conducted. This is the biggest missing link to judge the effectiveness of this work.

4. In reality, no classifier is completely individually fair. They are rather $\epsilon$-individually fair. What happens to the fairness of the proposed algorithms then?


[1] Michael Kearns, Aaron Roth, and Zhiwei Steven Wu. Meritocratic fairness for cross-population selection. In International Conference on Machine Learning - Volume 70, pp. 1828–1836. 2017.

[2] Thomas Kleine Buening, Meirav Segal, Debabrota Basu, Anne-Marie George, and Christos Dimitrakakis. "On Meritocracy in Optimal Set Selection." In Equity and Access in Algorithms, Mechanisms, and Optimization, pp. 1-14. 2022.

[3] Amanda Bower, Hamid Eftekhari, Mikhail Yurochkin, and Yuekai Sun. Individually fair rankings. In International Conference on Learning Representations. 2021.

---

### Decision · Action_Editors · 2023-08-03

**Recommendation:** Accept with minor revision

**Comment:**

The three reviewers have positive views on the manuscript. There are however several suggestions to improve the manuscript before it is published, particularly:
1) Strengthening discussion and explanation about ratio utility, and its relationship to the fair-utility maximising cohort selection problem under general individualistic utility and under any collective utility;
2) Positioning better the work with respect to existing literature in meritocratic fair selection/ranking. Explicitly mentioning that a meritocratic notion of fairness tries to preserve the order of the scores, whereas this work studies a stronger fairness criterion that preserves the distances between the scores. Cross-referencing with Buening et al. 2022 would be beneficial for the readers.

The matters mentioned above were addressed in the rebuttal; however, the manuscript has not been revised accordingly. A conclusion and discussion section would be appropriate to discuss those points. Additionally, there are several suggested modifications from the reviewers that need to be integrated.


**Audience:**

The area studied of individually fair and utility maximising cohort selection is significant and of interest to the TMLR community.

**Claims And Evidence:**

The manuscript proposes polynomial-time algorithms for fair and useful cohort selection with both linear and ratio utilities, in two different algorithmic models: offline and online settings. The proposed algorithms are based on a dependent rounding algorithm, which is a special case of rounding a fractional solution in a matroid polytope (a uniform matroid in their case). If a classifier that assigns scores to candidates is individually fair, the cohort/composition selection algorithms are shown to be either individually fair, or ε-individually fair. In the setting where candidates arrive in an online fashion, the algorithm allows putting candidates 'on hold', rather than requiring an immediate response.